# The toposiomerase IIIalpha-RMI1-RMI2 complex orients human Bloom's syndrome helicase for efficient disruption of D-loops

Gábor M. Harami [1,4,5 ✉], János Pálinkás[1,5], Yeonee Seol[2], Zoltán J. Kovács[1], Máté Gyimesi[1,3], Hajnalka Harami-Papp[1,4], Keir C. Neuman[2] & Mihály Kovács [1,3 ✉]

Homologous recombination (HR) is a ubiquitous and efficient process that serves the repair of severe forms of DNA damage and the generation of genetic diversity during meiosis. HR can proceed via multiple pathways with different outcomes that may aid or impair genome stability and faithful inheritance, underscoring the importance of HR quality control. Human Bloom's syndrome (BLM, RecQ family) helicase plays central roles in HR pathway selection and quality control via unexplored molecular mechanisms. Here we show that BLM's multi-domain structural architecture supports a balance between stabilization and disruption of displacement loops (D-loops), early HR intermediates that are key targets for HR regulation. We find that this balance is markedly shifted toward efficient D-loop disruption by the presence of BLM's interaction partners Topoisomerase IIIα-RMI1-RMI2, which have been shown to be involved in multiple steps of HR-based DNA repair. Our results point to a mechanism whereby BLM can differentially process D-loops and support HR control depending on cellular regulatory mechanisms.

[1] ELTE-MTA "Momentum" Motor Enzymology Research Group, Department of Biochemistry, Eötvös Loránd University, Pázmány P. s. 1/c, H-1117 Budapest, Hungary. [2] Laboratory of Single Molecule Biophysics, National Heart, Lung and Blood Institute, National Institutes of Health, Bethesda, Maryland 20892, USA. [3] MTA-ELTE Motor Pharmacology Research Group, Department of Biochemistry, Eötvös Loránd University, Pázmány P. s. 1/c, H-1117 Budapest, Hungary. [4] Present address: Laboratory of Single Molecule Biophysics, National Heart, Lung and Blood Institute, National Institutes of Health, Bethesda, Maryland 20892, USA. [5] These authors contributed equally: Gábor M. Harami, János Pálinkás. ✉email: gabor.harami@ttk.elte.hu; mihaly.kovacs@ttk.elte.hu

RecQ helicases are a conserved family of enzymes that play central roles in homologous recombination (HR) and several other nucleic acid metabolic processes in organisms ranging from bacteria to humans[1,2]. Mutations in three of the five human RecQ family members cause severe autosomal-recessive diseases[1]. Loss of function of Bloom's syndrome (BLM, RecQ family) helicase is associated with genomic instability, hyperrecombination, increased frequency of sister-chromatid exchange, growth deficiencies, increased cancer predisposition, female subfertility, and male infertility[3].

BLM helicase safeguards genome stability by playing key roles in homologous recombination (HR)-based repair of double-stranded (ds) DNA breaks (DSBs), the restart of stalled replication forks, and in the completion of chromosome replication[1]. Recent studies also indicate a crucial role for BLM in crossover regulation during meiosis[4,5]. A large body of evidence shows that BLM performs multiple DNA-processing activities supporting HR control and pathway selection[1,4,6].

BLM promotes dsDNA end resection, an early step initiating HR-based repair of dsDNA breaks, in complex with the MRN complex (MRE11, RAD50, and NBS1) and DNA exonucleases DNA2 or EXO1[7]. A role in resection has also been indicated for BTRR, a BLM-containing complex with topoisomerase IIIα, RMI1 (RecQ-mediated genome-instability protein 1), and RMI2, that forms specific interactions with replication protein A (RPA)[8,9]. Following resection, RAD51 recombinase catalyzes the invasion of the resulting single-stranded (ss) DNA overhang into a homologous intact DNA-duplex molecule, leading to the formation of displacement-loop (D-loop) structures (Fig. 1a). In addition to the prorecombination activity of BLM during resection, BLM has been shown to play multiple antirecombination roles. In vitro studies indicated that BLM can disrupt RAD51 recombinase nucleoprotein filaments, thereby inhibiting recombination[10,11]. However, recent studies revealed that BLM alone is unable to disrupt ATP-bound RAD51 filaments in vitro and, thus disruption is probably linked to the recombination-inactive, ADP-bound form[12] or additional, yet unexplored factors are required to disrupt active filaments. Nevertheless, an in vivo study highlighted that BLM regulates RAD51 filaments either by inhibiting their formation or facilitating their dissociation[10].

Disruption of nascent D-loops, a process in which BLM[11,13,14] and the BLM ortholog *Saccharomyces cerevisiae* (budding yeast) Sgs1–topoisomerase III–RMI (STR) complex was proposed to be involved[15,16], may serve to limit HR and/or mediate HR quality control by channeling HR into different pathways or by inhibiting illegitimate (nonallelic) recombination (IR) (Fig. 1a)[17,18]. Supporting this proposition, BLM's functions in HR control are reflected in the hyperrecombination and genome-instability phenotypes of BLM-deficient cells[1,3]. Accordingly, BLM and homologous *Escherichia coli* (*E. coli*) RecQ and Sgs1 helicases were shown to disrupt protein-free D-loop structures in vitro[11,13,14,19,20]. However, BLM or Sgs1 alone were unable to disrupt D-loops bound by stable, ATP-bound RAD51, despite being recruited to the site of strand invasion[12,20]. This indicates that BLM may act on D-loops on which RAD51 is destabilized or that other, yet unknown factors are involved in the processing of active RAD51-bound structures. The yeast topoisomerase III–RMI (TR) complex alone was shown to catalyze disruption of plasmid-based D-loops efficiently and also together with Sgs1 in vitro in the presence of recombinases and to disrupt nascent D-loops in vivo[15,20]. Strikingly, the catalytic activity of Sgs1 was not required for topoisomerase catalyzed D-loop disruption[20], but its presence was required for efficient removal of nascent D-loops in vivo[15].

D-loop disruption, when preceded by extension of the invading DNA segment by DNA synthesis, may promote synthesis-

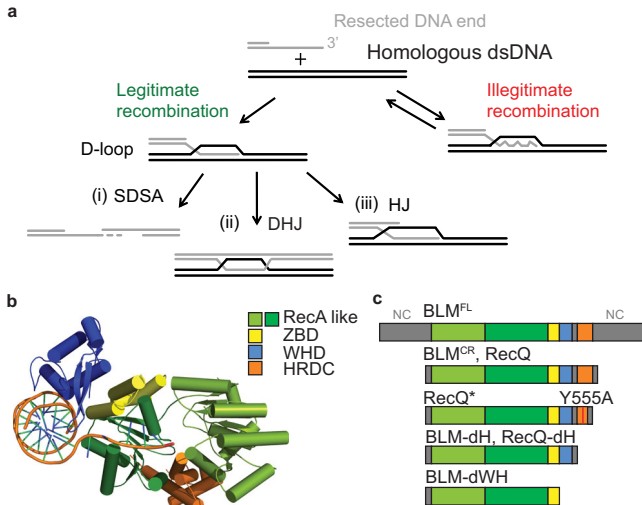

**Fig. 1 D-loop processing pathways and protein constructs used. a** During homologous recombination, a D-loop DNA structure (middle-left panel) is formed via the strand-exchange activity of recombinases, between the 3′-ssDNA overhang of a resected DNA molecule and an intact homologous dsDNA molecule. D-loops formed between nonallelic sequences can lead to illegitimate recombination (middle right panel); such events are thought to be reversed by disruption of the joint molecule (HR quality control). In case of allelic strand exchange (legitimate recombination), different modes of processing of the D-loop structure can channel recombination into different pathways. (i) In the presence of another resected DNA break end, disruption of D-loops extended by strand synthesis could lead to the synthesis-dependent strand-annealing (SDSA) pathway, resulting in noncrossover HR products. (ii) Annealing of the other resected DNA end to the D-loop could lead to the formation of a double Holliday junction (DHJ) structure, which can then be processed by DHJ resolution or dissolution. Structure-specific endonuclease-mediated resolution could lead to crossover or noncrossover HR products, whereas helicase- and topoisomerase-mediated DHJ dissolution generates noncrossover products (not shown). (iii) Migration of the branch of the D-loop structure away from the initial strand invasion (leftward in the figure) can stabilize the D-loop and possibly lead to the formation of a single HJ structure. This single HJ can be either resolved by nucleases or can be converted to a DHJ if the second DNA end is available. **b** Structure of a human BLM fragment comprising the helicase core (RecA-like and ZBD domains) and the WHD and HRDC domains in complex with a 3′-ssDNA-tailed dsDNA molecule (PDB code 4O3M)[35]. **c** Domain structure of BLM and RecQ constructs used in this study. RecQ constructs are identical to those used previously in Harami et al.[19]. "NC" denotes N- and C-terminal extensions specific to full-length BLM.

dependent strand annealing (SDSA) (Fig. 1a, pathway i), a crossover-free HR pathway[21]. RecQ helicases together with partner topoisomerases were implicated in the promotion of SDSA[21,22]. Intriguingly, BLM was able to extend a replication fork-like DNA structure (resembling also an extensible strand-invasion) in vitro along with DNA polymerase η, suggesting a function in stimulation of DNA-repair synthesis[11]. Moreover, BLM physically interacts with the p12 subunit of DNA polymerase δ[23], an enzyme implicated in D-loop extension during HR[24].

As an alternative to SDSA, D-loop stabilization and extension coupled to annealing of the second resected DNA end to the D-loop can lead to the formation of double Holliday junction structures (DHJ, Fig. 1a, pathway ii). These structures can be dissolved by the BTRR complex that exclusively generates non-crossover HR products, thereby avoiding potentially harmful consequences of crossover formation during somatic HR[25,26]. In

eukaryotic cells, similarly to what has been proposed to occur in bacteria[27,28], the D-loop structure could, in principle, also be stabilized by migration of the D-loop branch point, in a direction opposite to that leading to disruption (Fig. 1a, pathway iii). Such events would convert the D-loop into a single HJ structure that could be resolved by structure-specific endonucleases, dis-assembled, or converted to DHJ during further processing. Notably, most in vitro experiments testing enzymatic D-loop processing have been focused on disruption of the invading DNA segment, while the processing of other DNA regions in D-loop structures, e.g., branch migration leading to D-loop stabilization, has rarely been investigated.

Taken together, the above findings strongly suggest that the processing of D-loops is a crucial step in HR-pathway selection and quality control, and BLM along with the TRR complex (and in general BLM homologs together with homologs of topoi-somerase IIIα), is a central factor in determining the outcome of these processes, in addition to other D-loop processing helicases[4,29–31]. However, the molecular mechanisms by which BLM aids these processes remain elusive. Furthermore, whereas BLM forms a stable complex with TRR, it may also work inde-pendently of TRR in some cellular contexts. Thus, investigation of the effect of TRR on BLM's D-loop processing activities may shed light on previously unknown mechanisms.

BLM possesses a specific protein-domain architecture that is conserved in at least one RecQ homolog of each investigated organism (e.g., *E. coli* RecQ and yeast Sgs1) and appears to be important in supporting its multifaceted cellular roles[32–34]. This architecture comprises two RecA-like motor domains, required for ATP hydrolysis-driven ssDNA translocation (with 3′−5′ direc-tionality) and dsDNA unwinding activities, a protein structure-stabilizing zinc-binding domain (ZBD), and two auxiliary DNA-binding elements: the winged-helix domain (WHD) and the helicase-and-RNaseD-C-terminal (HRDC) domain (Fig. 1b and c). The WHD facilitates DNA unwinding and mediates protein–protein interactions[32,34–37]. The isolated HRDC domain of BLM binds weakly to ssDNA[38,39], contrary to the stronger ssDNA-binding affinity of *E. coli* RecQ HRDC[40]. The HRDC of BLM (and also that of *E. coli* RecQ) influences the ATPase activity of the protein via transient interactions with the motor core[19,35,41]. Recently, we demonstrated additional important mechanistic roles for the HRDC domain of *E. coli* RecQ, including inducing DNA geometry-dependent shuttling (local repetitive unwinding of DNA), and conferring a strong orientational preference for D-loop binding by the enzyme that results in efficient disruption of DNA-strand invasions[19]. In addition to the conserved domains, BLM possesses an N-terminal region involved in protein–protein interactions and oligomerization[42–44], and a C-terminal region containing the nuclear localization signal[45,46]. The latter two regions together are referred to as NC regions throughout this article.

Motivated by the above considerations, here we investigated the molecular mechanisms supporting the multifaceted HR-regulating functions of human BLM helicase and the BTRR complex by devising an analytical method to resolve all con-ceivable pathways of D-loop processing. In addition, we aimed to reveal the protein structural contributions to these activities using multiple engineered helicase constructs. We show that BLM alone balances D-loop processing pathways that could lead either to disruption or to stabilization of D-loops. Strikingly, this balance is strongly shifted toward D-loop disruption in the presence of the TRR complex. The intrinsic balance between the D-loop dis-ruption and stabilization activities of BLM and the effect of the TRR complex suggest that the regulation of these activities, through yet unexplored interactions with signaling or accessory factors, could be a key checkpoint for HR-pathway selection.

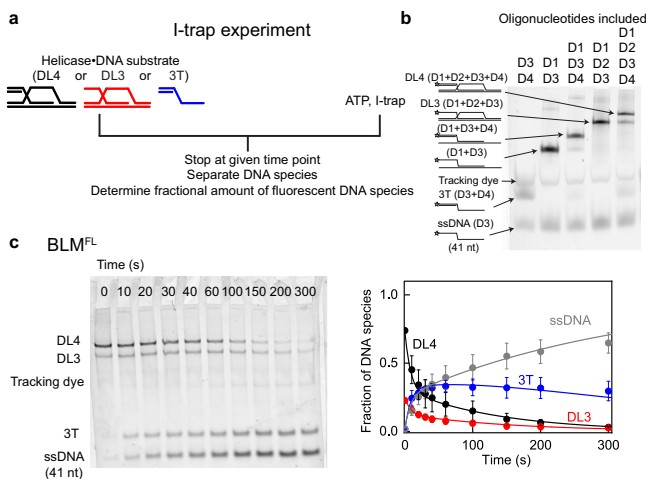

**Fig. 2 I-trap method to monitor D-loop processing. a** During I-trap D-loop processing experiments, preformed complexes of the given helicase construct with fluorescently-labeled D-loop-like (4-stranded DL4, or 3-stranded DL3) or 3′-tailed (3 T) DNA substrates (see Methods) were mixed with excess ATP and a large amount of unlabeled ssDNA trap strand (I-trap, homologous to the 3′ invading strand of DL4 and DL3)[19]. **b** Cy3 fluorometric image of a 12 w/v% native polyacrylamide gel electrophoretogram of DNA species (see Methods) involved in the I-trap DL4-disruption assays. Panel is reproduced from Harami et al.[19]. DL4 preparations contained fractions of DL3, 3 T, and ssDNA. **c** Example electrophoretogram of a DL4 unwinding experiment with full-length BLM (BLM^FL). Experiments were performed at least three times for individual protein constructs and showed similar kinetic profiles. Fractions of DNA species over time were determined by pixel densitometry. Means ± SEM are shown for the detected fractions of indicated DNA species at each time point for n = 3 independent experiments with BLM^FL (right panel). Lines correspond to global fits to the kinetic model shown in Fig. 3. Source data are provided as a Source Data file.

Elucidation of these regulation mechanisms will be key to understanding how eukaryotic cells achieve efficient and safe HR that supports the essential life processes of somatic genome maintenance and faithful chromosome inheritance during meiosis.

## Results

**Human BLM alone disrupts D-loops less efficiently than *E. coli* RecQ.** Previously, we developed a D-loop disruption enzyme kinetic assay using protein-free, oligonucleotide-based DNA structures to dissect D-loop disruption pathways of *E. coli* RecQ helicase (referred to as RecQ throughout this article). Using this assay, we demonstrated that RecQ efficiently disrupts D-loop-like structures by removing strand invasions, and that this activity is mediated by its HRDC domain[19]. BLM was also shown to unwind D-loop structures[11,13,14,32]. However, BLM was also shown to facilitate DNA-repair synthesis in vitro[11], which could lead to stabilization of D-loop structures in an in vivo context, suggesting differences in D-loop processing by BLM and RecQ.

To explore BLM's D-loop processing pathways and to test the role of BLM-domain architecture in D-loop processing, first, we utilized our previously developed kinetic assay (Fig. 2). Briefly, in separate experiments we monitored processing of Cy3-labeled four-stranded and three-stranded D-loop-like DNA structures (DL4 and DL3, respectively; Fig. 2b, see "Methods" for oligonucleotide composition and sequences) along with a 3′ tailed dsDNA substrate (3T, part of the DL4 structure, Fig. 2b), in the presence of various BLM constructs (Fig. 1c, Supplementary Fig. 1), ATP, and an ssDNA trap strand (I-trap, unlabeled version

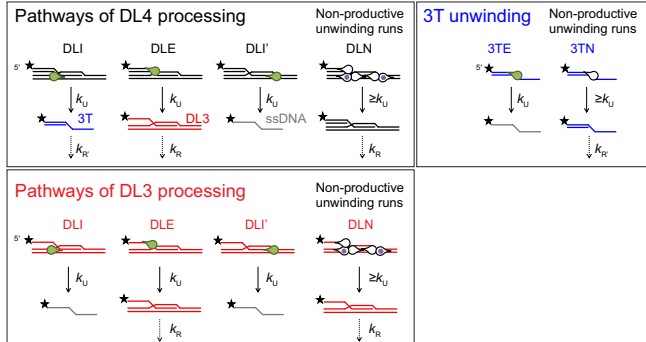

**Fig. 3 Kinetic pathways of processing of DL4, DL3, and 3T DNA substrates.** The model has been described previously in Harami et al.[19] and is summarized as follows. Fluorescently-labeled D-loop-like (4-stranded DL4, or 3-stranded DL3), 3′-tailed (3T) dsDNA, and ssDNA structures are colored black, red, blue, and gray, respectively. DNA-bound helicases are shown as green drops pointing toward the direction of unwinding (3′–5′ on the tracked DNA strand). Black stars represent the Cy3 fluorescent label at the 5′ end of the invading DNA strand. Enzyme–DNA complexes are initially distributed into different configurations (DLI, DLE, DLI′, DLN, 3TE, and 3TN). For DLI, enzyme–DL4 or DL3 complexes are oriented for disruption of the invasion, producing 3T from DL4 or ssDNA from DL3. For DLE, the enzyme is oriented "outward" from the invasion and will produce DL3 from DL4—or in the case of DL3 substrate, leave the substrate intact. For DLI′, the enzyme tracks the invading strand starting from its 3′ end, thus producing ssDNA from either DL4 or DL3. DLN and 3TN represent all nonproductive unwinding runs starting from any possible enzyme–DNA configuration (white drops). Importantly, DLN also includes the fraction of proteins bound in orientations favoring unwinding of the D-loop dsDNA arms (indicated by purple dots). Even successful unwinding of these arms will be nonproductive due to rapid reannealing of the unwound DNA strand. Unwinding (occurring at rate constant $k_U$) leads to the indicated DNA products. Slow rebinding of the enzyme to these DNA products (inhibited by excess I-trap ssDNA trap strand, occurring at $k_R$ for DL4 and DL3, or $k_{R'}$ for 3T) leads to reformation of enzyme–DNA configurations (with the same distribution as for the initial complexes; reformation and redistribution together are indicated by downward arrows labeled $k_R$ and $k_{R'}$). These pathways and associated rates form the basis of the kinetic model that is used to fit the data, e.g., in Figs. 2c, 4.

of the labeled invading strand; Fig. 2c, additional control experiments are shown in Supplementary Fig. 2). In addition to full-length BLM helicase (BLM^FL, comprising the conserved RecQ domains and the NC regions), we used a well-characterized monomeric BLM construct possessing a domain structure identical to that of RecQ (amino acids (aa) 642–1290, denoted as BLM^CR, lacking the NC regions)[33], as well as HRDC-truncated (BLM-dH, aa 642–1191) and WHD-HRDC-truncated (BLM-dWH, aa 642–1077) BLM constructs[32] (Fig. 1c, Supplementary Fig. 1).

Our previously devised kinetic model (Fig. 3)[19] was globally fitted to the DL4, DL3, and 3T unwinding reactions for each BLM construct (Fig. 4a) to determine the initial partitioning of enzyme–substrate complexes and the rate constants of the model (Fig. 4b, Supplementary Table 1). The model incorporates multiple D-loop processing pathways dictated by different enzyme–DNA-binding configurations and the 3′–5′ unwinding directionality of the helicase. Modeling indicated that BLM constructs are generally less efficient in disrupting invasions than RecQ (Fig. 4 and ref. [19]). BLM^FL disrupts DL4 and DL3 via unwinding the strand invasion (DLI pathway, cf. Fig. 3) with the highest probability (Fig. 4b, Supplementary Table 1). Nevertheless, the fraction of this pathway is markedly lower, and the

fractions of the DLI′ pathway and nonproductive unwinding runs (DLN) (Fig. 3) are higher, compared with those for RecQ[19]. These results suggest that BLM lacks the strong bias toward invasion disruption that was observed for RecQ[19]. BLM^FL unwound the 3T DNA structure (Fig. 4a) almost as efficiently as RecQ, and the binding affinity of BLM^FL to the investigated DNA structures (Supplementary Fig. 3, Supplementary Table 1) was similar to that of RecQ (cf. ref. [19]), indicating that the differences in pathway partitioning for BLM and RecQ do not result from attenuated dsDNA unwinding or DNA-binding activities of BLM.

**The accessory domains of BLM maintain balance between D-loop disruption and stabilization.** The large fraction of nonproductive runs (DLN pathway) observed for BLM^FL (and other constructs) in D-loop unwinding experiments (Fig. 4b) can occur either because the enzyme unwinds one of the dsDNA arms flanking the invasion (which will then rapidly reanneal), or because the enzyme prematurely terminates unwinding initiated from any of the possible binding configurations (Fig. 3). To test these possibilities, we performed DL4-processing experiments based on a design in which either a fluorescein-labeled ssDNA strand that can anneal to the "bottom" DNA strands of the dsDNA arm upstream of the invasion (L-trap), or one that can anneal downstream of the invasion (R-trap) was added in large excess (compared with DL4 concentration) along with the I-trap and ATP (Fig. 5). These strands will only anneal if the original dsDNA arms were unwound by the helicase. Using this assay, we investigated the activity of BLM and RecQ constructs (Figs. 1c, 6–7, Supplementary Fig. 1), with the exception of BLM-dWH and RecQ-dWH as these proteins failed to effectively unwind DL4 or DL3 (Fig. 4a and ref. [19]).

Importantly, using the L-trap, unwinding of the "left-side" dsDNA arm; and using the R-trap, unwinding of the "right-side" dsDNA arm can be separately monitored, as binding of the fluorescent traps inhibits reformation of the given dsDNA arm (Fig. 5). By monitoring the Cy3 signal that reports the presence of the invading strand, and the fluorescein signal that reports the presence of the L- or R-trap strand in the given DNA species (Supplementary Figs. 4–5), we can now detect and distinguish among all conceivable D-loop unwinding orientations (Supplementary Fig. 7) that comprise the nonproductive DLN fraction in the previous experiments (Fig. 3). In the presence of L-trap, a D-loop binding orientation that leads to unwinding of the left dsDNA arm (DLL pathway, Fig. 5a, Supplementary Fig. 7) will generate the five-stranded DL5L DNA structure from DL4 (Fig. 5a, Supplementary Figs. 4, 6a), whereas in the presence of R-trap, a D-loop binding orientation that leads to unwinding of the right arm (DLR pathway, Fig. 5b, Supplementary Fig. 7) will generate the five-stranded DL5R structure (Fig. 5b, Supplementary Figs. 5, 6b).

In L-trap unwinding experiments (Supplementary Fig. 4), in addition to bands that were observed in previous experiments where the Cy3 fluorescence was monitored (Fig. 2c), a band migrating slower than DL4 appeared rapidly and then disappeared slowly over time. This low-mobility band was also observed when the fluorescein signal was monitored (Supplementary Fig. 4). Control experiments (Supplementary Fig. 6a) revealed that this band corresponds to the expected five-stranded DNA structure (DL5L) in which both the Cy3-labeled invading strand and the fluorescein-labeled L-trap strands are present simultaneously (Fig. 5a). DL5L disappeared over time, likely due to slow rebinding of the helicase to DL5L from the trap strands followed by unwinding.

In R-trap experiments (Supplementary Fig. 5), we never observed a band corresponding to the five-stranded DL5R

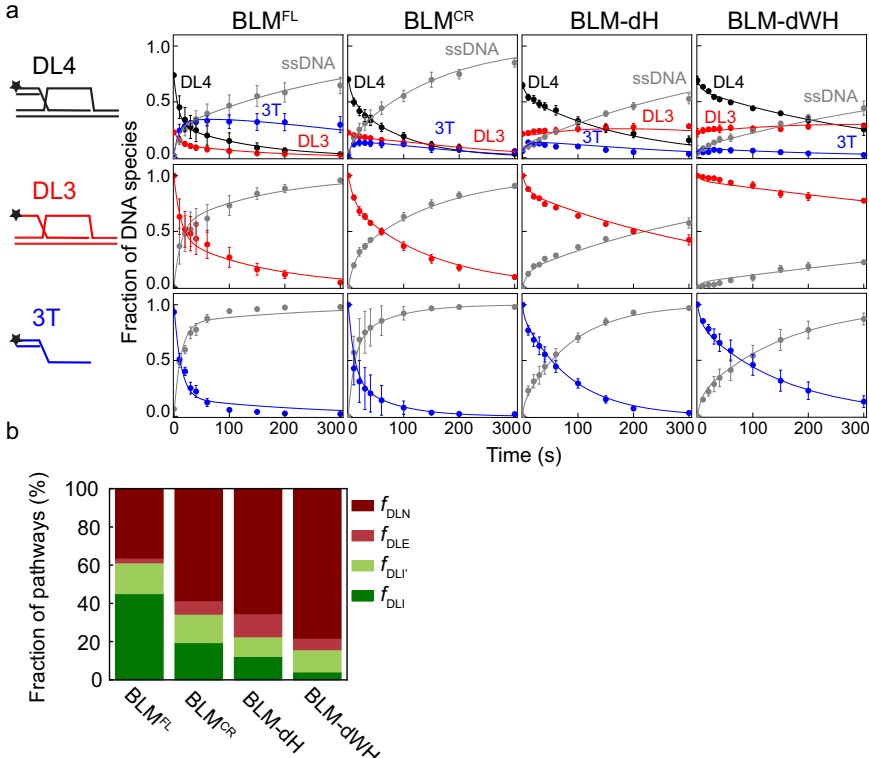

**Fig. 4 BLM alone does not effectively disrupt D-loop invasions. a** Kinetic profiles of DL4, DL3, and 3T unwinding by BLM$^{FL}$, BLM$^{CR}$, BLM-dH, and BLM-dWH. Fractions of DNA species were determined by pixel densitometry from I-trap D-loop unwinding experiments (a typical result is shown for BLM$^{FL}$ in Fig. 2c). Color coding of fractions of DNA species is as in Fig. 3 (DL4 (4-stranded D-loop-like structure): black, DL3 (3-stranded D-loop-like structure): red, 3T (3′-tailed dsDNA): blue, ssDNA: gray). Solid lines show global best fits of the model shown in Fig. 3 to all unwinding data (DL4, DL3, and 3T) of each enzyme variant. DNA substrates and DNA species observed during the reaction are described in Fig. 2b and Methods. For all BLM constructs, the disappearance of DL4 during the unwinding time courses was accompanied by a transient accumulation of 3T, eventually converted to ssDNA. Similar kinetic profiles were obtained for RecQ constructs[19]. Means ± SEM (n = 3) are shown on all panels for the detected fractions of DNA species at each time point determined from independent experiments with individual protein constructs. **b** Distributions of enzyme–DL (DL4 or DL3) configurations obtained from global fits shown in panel **a** (determined parameters are listed in Supplementary Table 1). Deletion of the NC region (BLM$^{CR}$ construct) increased the fraction of both the DLE and DLN pathways (cf. Fig. 3), however, the NC regions are dispensable for strong binding to DL4, DL3, or 3T (Supplementary Fig. 3b and c). Further deletion of the HRDC domain (BLM-dH) slightly increased the fraction of the DLE and DLN pathways, meanwhile, it lowered the fraction of the DLI pathway (compared with BLM$^{CR}$). These results suggest that the HRDC domain plays a minor role in promoting specific BLM–D-loop interactions, in line with DL4, and DL3 binding measurements (Supplementary Fig. 3d). Deletion of the BLM WHD and HRDC domains (BLM-dWH) further increased the fraction of the DLN pathway, whereas it decreased the DLI fraction and decreased the unwinding efficiency of the 3T substrate (panel a), indicating a moderately lowered unwinding processivity compared with that of other constructs. Source data are provided as a Source Data file.

structure (Fig. 5b) for any of the investigated helicase constructs. Based on control experiments, the DL5R structure is detectable via the R-trap DNA strands, albeit the fainter bands in DNA species where the complementary strand (D2) and the R-trap are present indicate that the detection sensitivity is lower compared with L-trap experiments (Supplementary Fig. 6). Nevertheless, the R-trap results indicate that neither helicase construct prefers the DLR D-loop processing pathway (Fig. 5b).

Importantly, no changes in the fraction of the DNA species originally present in our DL4 preparation (DL4, DL3, 3T, and ssDNA) and no appearance of five-stranded species (DL5L or DL5R) were observed in L-trap or R-trap control experiments performed without ATP (hence, in the absence of helicase activity, Supplementary Fig. 8a). In addition, the presence of L-trap or R-trap does not influence the kinetics of helicase activity (Supplementary Fig. 8b).

To determine the partitioning among different D-loop processing pathways, we performed global kinetic fitting to DL4 data obtained with L- or R-trap and DL3 and 3T data obtained with I-trap (Figs. 6–7, Supplementary Fig. 9) using an extended model (Supplementary Fig. 7, cf. that was used in Fig. 3) that

included the DLL and DLR pathways. Analysis of the data with this model (described in Methods) confirmed that BLM$^{FL}$ is not as effective in D-loop disruption as RecQ (Fig. 7d, Supplementary Table 2), in agreement with the I-trap-only data (Figs. 3, 4, 6c, 7c). Importantly, the extended model indicates that this difference is due to the different D-loop processing pathway preference of BLM (compared with that of RecQ, Fig. 7d), and not to differences in enzyme processivity. This finding is also supported by the mean unwinding run length of RecQ and BLM[12,19] that exceeds the length of dsDNA regions comprising our D-loop structure.

Strikingly, BLM maintains a balance between the DLI plus DLI′ and DLL pathways, whereas RecQ has a clear bias toward the DLI pathway (Fig. 7d, see pathways in Supplementary Fig. 7). In the cellular context, the DLI and DLI′ pathways would lead to elimination of the strand invasion; in contrast, the DLL pathway could lead to stabilization of the strand invasion (Fig. 5a, Supplementary Fig. 7). The BLM NC regions moderately influence the partitioning among pathways: in the absence of the NC regions (i.e., in BLM$^{CR}$, compared with BLM$^{FL}$), the fraction of the DLI plus DLI′ pathways decreased slightly (by

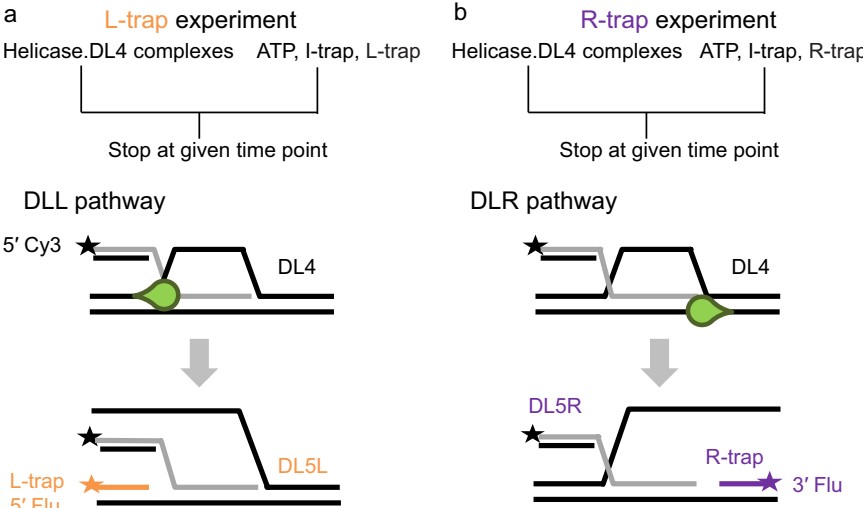

**Fig. 5 Schematic representation of L-trap and R-trap experiments to dissect and quantify all conceivable D-loop processing pathways. a**, **b** Mixing fluorescein-labeled ssDNA trap strand (**a**, L-trap (orange) or **b**, R-trap (purple)) along with the I-trap and ATP with the helicase (green drop)—DL4 complexes (gray invading strand; black star: Cy3 label) during D-loop processing experiments allows resolution of the previously unobservable unwinding of the dsDNA "arms" of the D-loop structure (see also the orientations labeled with purple dots in the DLN pathway shown in Fig. 3) in addition to previously observable pathways (Fig. 3). In L-trap experiments, unwinding of the dsDNA arm upstream of the invasion will lead to the formation of the five-stranded double-labeled DL5L structure, whereas in R-trap experiments, the unwinding of the dsDNA downstream of the invasion will lead to the formation of the five-stranded double-labeled DL5R structure, due to the complementary binding sites for the labeled trap strands. DNA sequences are described in Methods.

about 1.4-fold), whereas the fraction of the nonproductive (DLN) pathway increases concomitantly (by about 2.6-fold) and the fraction of DLL remained practically unchanged (Fig. 7d, Supplementary Table 2). The deletion of the HRDC domain (in BLM-dH) further decreased the DLI plus DLI′ fractions (by about 2.0-fold compared with $BLM^{CR}$) and increased the DLN fraction by about 2.3-fold compared with $BLM^{CR}$ and the DLL fraction remained practically unchanged again (Fig. 7d, Supplementary Table 2). Importantly, these results indicate that the NC region and the HRDC domain are both required for processive invasion disruption via the DLI (plus DLI′) pathways, but not for unwinding occurring in the DLL pathway. In contrast, deletion of the RecQ HRDC domain (RecQ-dH) decreased the DLI fraction by about 3-fold and increased the DLL fraction about 5-fold (Fig. 7d, Supplementary Table 2). In addition, the DLN fraction increased about 4-fold in RecQ-dH compared with that in RecQ. Practically, $BLM^{CR}$ behaved similar to RecQ-dH, indicating differential roles of HRDC domains in the two proteins and/or the requirement of other C-terminal elements in BLM for proper HRDC functions. A RecQ construct in which the Y555A substitution abolishes the ssDNA binding ability of the HRDC domain (RecQ*)[40] retained the bias toward the DLI pathway; however, the DLI fraction was slightly decreased (by a factor of about 1.2), whereas the DLL fraction increased (by about 2.7-fold) without changes in the DLN fraction (Fig. 7d, Supplementary Table 2). These results indicate that the presence of the RecQ HRDC domain, but not its ssDNA binding ability, is required for the bias toward the DLI pathway. The unchanged DLN fraction indicates that the HRDC domain's ssDNA binding ability does not influence the processivity in such a way that would manifest during D-loop processing.

**The TRR complex directs BLM's D-loop processing activity toward invasion disruption.** Next, we tested the effect of the TRR complex on D-loop processing by BLM. In control experiments, we found that the TRR complex binds strongly to the investigated DNA structures (Supplementary Fig. 10a) with the

affinity order of DL3 > ssDNA (ss54)>DL4. In addition, we observed that TRR alone is able to disrupt a fraction of the DL4 structure accompanied by the formation of the 3T structure in a TRR concentration-dependent, but ATP-independent manner (Supplementary Fig. 10b). Interestingly, the amount of DL3 did not change markedly upon addition of TRR, and TRR was not able to disrupt the ssDNA-overhanged 3T dsDNA structure. Since TRR binding to DNA structures and TRR-mediated D-loop disruption reaches quasi-saturation at TRR concentrations around 1 μM, we decided to use a 1.2 μM TRR concentration in the kinetic assays monitoring the D-loop disruption activity of BLM (leading to a twelve-times molar excess of TRR over BLM). During the preincubation phase of the assay, where TRR and BLM are incubated with the D-loop structure in the absence of ATP, TRR-facilitated D-loop disruption reaches an equilibrium and this process is not influenced by $BLM^{FL}$ (Supplementary Fig. 10c and d). Thus, TRR's activity alone does not influence the fractionation of DNA species during the timeframe over which BLM's unwinding activity is monitored after addition of ATP. Together, these control experiments confirmed that D-loop processing by BLM can be monitored in the presence of the TRR complex even in the μM concentration regime.

Next, we measured the effect of TRR on the D-loop disruption and unwinding activities of $BLM^{FL}$ and $BLM^{CR}$ using the DL4, DL3, and 3T DNA structures in separate L- and R-trap (R-trap with $BLM^{FL}$) experiments. Also, as a control, we used *E. coli* RecQ that does not interact with TRR. Assays were repeated also without addition of TRR to obtain specific controls for these experiments (independent of TRR-free results described above). In this way, the effects of TRR can be directly dissected by comparing the results of parallel experiments performed with or without TRR.

Generally, investigated proteins exhibited similar behavior to that observed previously in L-trap experiments (Supplementary Fig. 4). In case of $BLM^{FL}$, the formation and concomitant disruption of DL5L (i.e., the presence of the DLL pathway) was apparent (Supplementary Fig. 11a). $BLM^{CR}$ showed altered DL5L

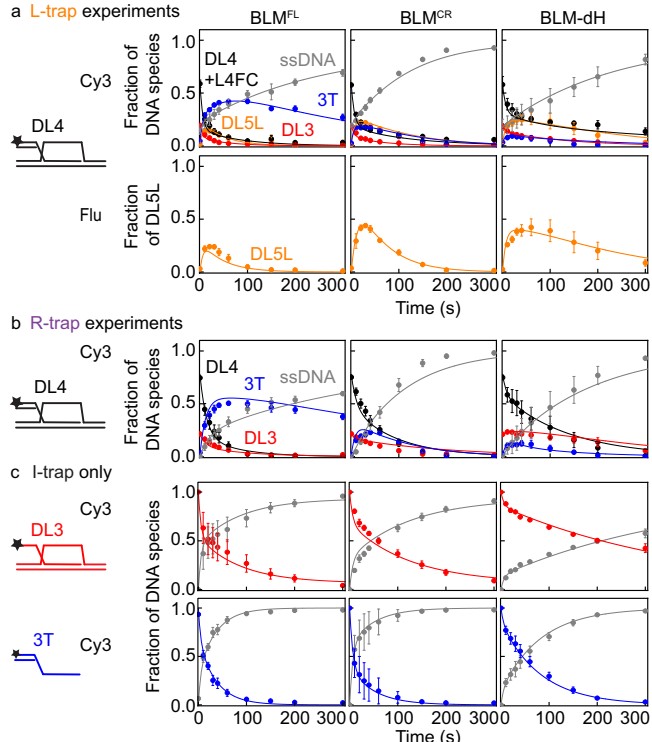

**Fig. 6 Kinetics of D-loop processing by BLM constructs. a**, **b** Kinetic profiles of DL4 unwinding by BLM$^{FL}$, BLM$^{CR}$, and BLM-dH in the presence of (**a**) L-trap or (**b**) R-trap. The fraction of DL5L compared with initial DL4 was determined from the fluorescein signal (cf. Supplementary Fig. 4) by dividing the measured pixel density by the density of 30 nM L-trap multiplied with the initial fraction of DL4 determined from Cy3 scanning of the same gel. In R-trap experiments (Supplementary Fig. 5), we did not observe the formation of DL5R for any BLM construct. **c** Kinetic profiles of DL3 and 3T unwinding obtained in I-trap-only experiments. Data shown are identical to those in Fig. 4a (used for both types of analysis). Color coding of DNA species is as in Figs. 3, 5, and Supplementary Fig. 7: DL5L (five-stranded structure): orange, DL4 (4-stranded D-loop-like structure): black, DL3 (3 stranded D-loop like structure): red, 3T (3′-tailed dsDNA): blue, and ssDNA: gray. Means ± SEM ($n = 3$) are shown on all panels for the detected fractions of DNA species at each time point determined from independent experiments with individual protein constructs. Solid lines show global best-fits of the model described in Supplementary Fig. 7 to all unwinding data (DL4, DL3, and 3T) of each enzyme variant. Source data are provided as a Source Data file.

kinetics compared with the full-length protein, whereas DL5L formation was not observed with RecQ (Supplementary Fig. 11b). Reaction rates were detectably faster for all proteins compared with the earlier experiments, possibly arising from the use of new batches of reagents; however, these parameters did not influence any of the conclusions drawn below.

In L-trap experiments, the presence of the TRR complex markedly altered the behavior of BLM$^{FL}$, most apparent via the absence of DL5L, indicating that TRR altered the relative probability of D-loop processing pathways (Supplementary Figs. 7, 11a). In addition, D-loop processing kinetics were detectably faster compared with results in the absence of TRR, reflected in the faster disappearance of DL4 and DL3 species. DL4 control experiments with increased ssDNA trap-strand concentrations indicated that these effects are not caused by alterations in single-round DNA unwinding conditions due to changes in the concentration of ssDNA trap strands in the presence of TRR (Supplementary Fig. 12a). In R-trap experiments, the formation

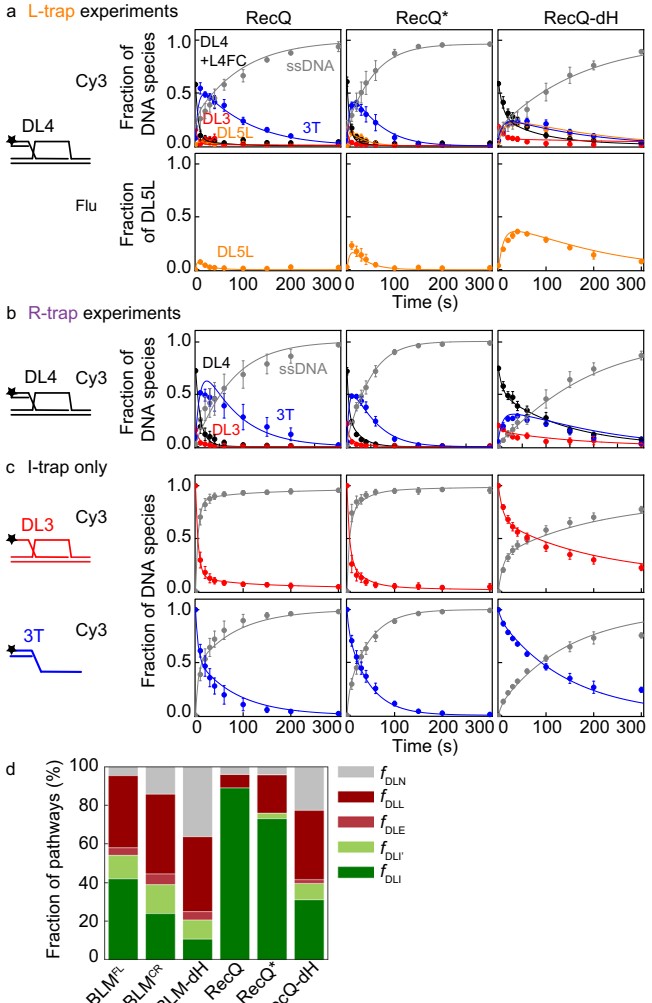

**Fig. 7 Kinetics of D-loop processing by RecQ constructs. a**, **b** Kinetic profiles of DL4 unwinding by RecQ, RecQ*, and RecQ-dH in the presence of (**a**) L-trap or (**b**) R-trap. The fraction of DL5L compared with initial DL4 was determined as in Fig. 6a. **c** Kinetic profiles of DL3 and 3T unwinding obtained in I-trap-only experiments. Data shown were replotted from Harami et al.[19]. Color coding of DNA species is as in Figs. 3, 5, and Supplementary Fig. 7: DL5L (five-stranded structure): orange, DL4 (4-stranded D-loop-like structure): black, DL3 (3-stranded D-loop-like structure): red, 3T (3′-tailed dsDNA): blue, and ssDNA: gray. Solid lines show global best-fits of the model described in Supplementary Fig. 7 to all unwinding data (DL4, DL3 and 3T) of each enzyme variant. Means ± SEM ($n = 3$) are shown on all panels for the detected fractions of DNA species at each time point determined from independent experiments with individual protein constructs. **d** Distributions of enzyme–DL (DL4 or DL3) configurations (Fig. 3, Supplementary Fig. 7) obtained from global fits to the data shown in Figs. 6 and 7a–c. Configurations leading to D-loop disruption are colored with green shades, whereas those leading to D-loop stabilization in an in vivo context are colored with shades of red. Nonproductive fractions are colored gray. Determined parameters are listed in Supplementary Table 2. Source data are provided as a Source Data file.

of DL5R, i.e., unwinding of the right dsDNA arm was not observed either without TRR (in line with previous results) or with TRR (Supplementary Fig. 12b). Thus, we conclude that TRR does not increase the preference of BLM for this pathway (see Supplementary Fig. 7 for depiction of pathways).

In general, the activity of BLM$^{CR}$ was not influenced by TRR as markedly as that of the full-length protein: DL5L accumulation

decreased moderately, but was still observable and the kinetics of DL4 and DL3 disruption appeared unaltered (Supplementary Fig. 11b). Strikingly, the D-loop processing activity of RecQ was markedly inhibited by TRR: unwinding kinetics were slowed for all DNA species (Supplementary Fig. 11b). Importantly, these results indicate that TRR binds to DL3 and DL4, in line with binding experiments (Supplementary Fig. 10), and in the absence of interactions between the helicase and TRR, DNA binding by TRR impedes helicase-driven DNA unwinding. Furthermore, these results highlight that the different TRR-induced changes in the behavior of different BLM constructs are caused by differing extents of protein–protein interactions and not due to other possible artificial effects.

Global fitting of the results with our extended model (Fig. 8, Supplementary Fig. 13, Supplementary Table 3) reflected increased unwinding and rebinding rates for all proteins in the absence of TRR compared with the earlier results (Supplementary Tables 2 and 3). Importantly, however, the fractionation of D-loop processing pathways remained similar (Fig. 7d and Fig. 8b), highlighting the robustness of our method for prediction of D-loop processing pathway fractionation. As seen previously, BLM$^{FL}$ maintains a balance between pathways, leading to D-loop disruption (DLI + DLI′) or stabilization (DLL); the absence of the NC regions in BLM$^{CR}$ slightly decreases the fraction of D-loop disruption pathways, whereas RecQ has a preference for D-loop disruption (Fig. 8b, Supplementary Table 3). However, in the presence of the TRR complex, BLM$^{FL}$'s preference for D-loop disruption (DLI plus DLI′) increased markedly (by about 1.6-fold), whereas the DLL pathway was much less favored (by about 11-fold). In addition, the DNA unwinding rate was predicted to be enhanced by TRR by about 2-fold (Supplementary Table 3). Control experiments performed at higher time resolution confirmed that TRR increases the D-loop unwinding activity and, importantly, revealed in a model-independent manner that the absence of the DLL pathway in the presence of TRR is not due to the increased reaction rates (Supplementary Fig. 14). In contrast to the full-length BLM protein, in the presence of TRR, BLM$^{CR}$'s participation in the D-loop disruption pathways (DLI + DLI′) and the DLL pathway was lowered accompanied by a predicted large increase in the DLE pathway fraction (by about 7-fold). In contrast to BLM$^{FL}$, the presence of TRR did not enhance the apparent unwinding rate of BLM$^{CR}$ (Supplementary Table 3). *E. coli* RecQ retained its preference for D-loop disruption pathways in the presence of TRR, however, both the unwinding and rebinding rates were markedly lowered (Supplementary Table 3).

## Discussion

Our D-loop unwinding experiments (Figs. 2–8, Supplementary Figs. 1–8) revealed a striking property of BLM: the helicase maintains a balance between binding to oligonucleotide-based, protein-free D-loop-like structures (mimicking a 3′-strand invasion) in orientations that will lead to the disruption of the strand invasion and those that could, in an in vivo context, lead to the stabilization of the D-loop structure (DLL pathway, Fig. 9). Previous multiround D-loop unwinding experiments also indicated unwinding of the dsDNA arms of similar oligonucleotide-based structures; however, this activity was not detectable using plasmid-based D-loops[13].

The NC regions and the HRDC domain moderately influence D-loop processing (Figs. 6 and 7d, Supplementary Tables 1–2) without affecting the apparent D-loop binding affinity (Supplementary Fig. 3, Supplementary Table 1). These domains are required for effective disruption of a strand invasion (DLI pathway, Fig. 7d). Interestingly, however, these domains do not

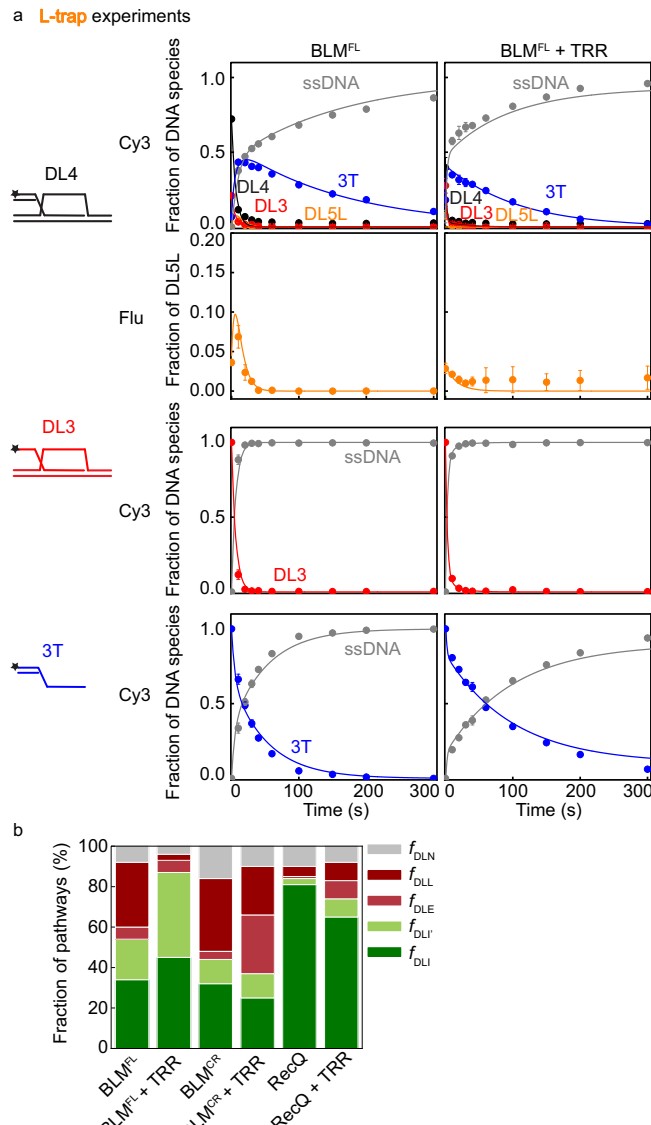

**Fig. 8 Effect of TRR on the D-loop processing activities of BLM and RecQ constructs. a** Kinetic profiles of DL4, DL3, and 3T unwinding by BLM$^{FL}$ in the absence and presence of 1.2 μM TRR. Additional kinetic profiles for other protein constructs are shown in Supplementary Fig. 13. Solid lines show global best-fits of the model described in Supplementary Fig. 7 to all unwinding data (DL4, DL3, and 3T) of BLM$^{FL}$ with and without TRR. Means ± SEM are shown on all panels for the detected fractions of DNA species at each time point determined from independent experiments (*n* = 3 for DL4 experiments, *n* = 2 for 3T and DL3 experiments) with individual protein constructs. **b** Distributions of enzyme–DL (DL4 or DL3) configurations resulting from global fits shown in panel a and Supplementary Fig 13. Configurations leading to D-loop disruption are colored with green shades, whereas those leading to D-loop stabilization in an in vivo context are colored with shades of red. Determined parameters are listed in Supplementary Table 3. Source data are provided as a Source Data file.

markedly influence the probability of unwinding of the "left" D-loop dsDNA arm and thus the effectiveness of D-loop stabilization (Figs. 7d and 9, Supplementary Table 2). Importantly, these effects are not caused by different oligomerization states of the constructs: whereas the NC region has been implicated in BLM oligomerization in the absence of DNA and ATP[42], we showed that the NC regions do not induce significant

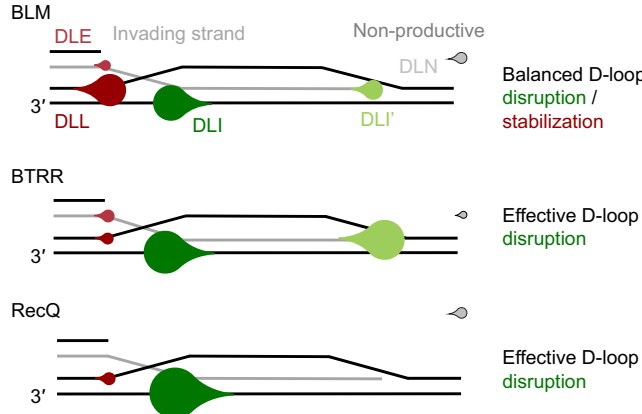

**Fig. 9 D-loop processing pathway preference of BLM, BTRR and RecQ.** Graphical presentation of the preference of binding orientations and the fraction of nonproductive unwinding events for the given helicase constructs. The tip of the droplets indicates the direction of unwinding. Relative sizes of droplets are proportional to the fractions shown in Fig. 7d and Fig. 8b. Configurations leading to D-loop disruption are colored with green shades, whereas those leading to D-loop stabilization in an in vivo context are colored with shades of red. Nonproductive events are represented by gray droplets.

oligomerization of BLM on the D-loop structure (both BLM[FL] and the truncated constructs bind the D-loop-like structure with close to 1:1 stoichiometry)[47].

Previously, and also in this study, we showed that RecQ dominantly binds to D-loop structures in an orientation that leads to disruption of strand invasions (DLI pathway in Figs. 3, 7d and 9, Supplementary Fig. 7, Supplementary Table 2)[19]. The binding orientation preference of RecQ is brought about by the presence of its HRDC domain, but the HRDC domain's moderate ssDNA binding affinity is not required for this binding orientation preference and, hence, invasion disruption (Figs. 7d and 9, Supplementary Table 2). The WHD is required for strong binding to D-loop-like structures and for processive D-loop disruption in both BLM (Fig. 4, Supplementary Fig. 3e, Supplementary Table 1) and RecQ[19]. The observed differences in D-loop processing between RecQ and BLM may originate from different interaction properties of their homologous protein domains. Whereas the structures of BLM and RecQ HRDC and WH domains are overall highly similar, their amino acid sequences are not conserved and small differences, such as the insertion of an acidic region into the BLM HRDC domain (absent in RecQ) indicate functional fine-tuning, differentiating it from RecQ[35,36,38,39,48,49]. Alternatively, the HRDC domain of BLM may require other parts of the C-terminal elements to function properly.

Surprisingly, we found that the TRR complex enhances BLM's D-loop unwinding activity and orients the helicase towards efficient D-loop disruption (Figs. 8 and 9, Supplementary Table 3). Previous experiments showed that the TRR complex works together with BLM to dissolve double Holliday junctions during late stages of HR-based DNA repair, and BLM was found to stimulate the activity of topoisomerase IIIα[26,44,50]. Abolishing the interaction between BLM and topoisomerase IIIα through deletion of the NC regions led to the loss of the stimulatory effect. In line with this observation, in our experiments the stimulatory effect of TRR on BLM's activity was lost in the absence of BLM's NC regions. However, while interactions with topoisomerase IIIα and RMI1 were mapped to BLM's N-terminal region[44,51], residual interactions between BLM[CR] and RMI1-2 (and TRR) may allow unwinding of DNA regions initially bound by TRR. In contrast, *E. coli* RecQ, which does not interact with TRR, was

unable to process TRR-bound D-loops efficiently (Supplementary Fig. 13).

The observed propensity of BLM to maintain a balance between D-loop disruption and stabilization and the propensity of TRR to drive BLM into pathways leading to efficient D-loop disruption (Figs. 8b and 9) may provide mechanistic clues for the multifaceted, often antagonistic in vivo roles of BLM.

BLM is involved in a multitude of DNA metabolic processes: it was shown to be a key player in the initiation and quality control of HR-based DNA repair, it is involved in telomere replication, the reinitialization of stalled replication forks, the removal of late-replication intermediates, and in the progression of meiotic recombination[1]. Multiple lines of cellular evidence show that, in the absence of BLM in humans or Sgs1 in yeast, the frequency of aberrant joint molecules increases, indicating a role for these enzymes in the regulation and quality control of DNA-strand invasions, among other functions[22,31,52–54]. Accordingly, BLM and Sgs1 were found to disrupt protein-free D-loops in vitro[11,14,20]. However, BLM was only able to disrupt D-loop structures when RAD51 was inactivated[11,12] and Sgs1 alone also failed to disrupt D-loops formed with active RAD51[20]. In single-molecule experiments, the BTRR complex was unable to disrupt ssDNA-bound RAD51 filaments, but BLM was recruited to the heteroduplex joints of D-loops[12]. These results indicate that helicase-mediated D-loop disruption is likely limited to contexts where RAD51 is already inactivated/removed and/or additional, yet unknown factors can enhance the D-loop disruption activity. Nevertheless, another study showed that the human TRR complex can disrupt plasmid-based D-loops without bound proteins (in line with our results) or bound by active RAD51[20]. This study also revealed that the yeast TR complex can efficiently disrupt RAD51-bound D-loop structures, and interestingly, the helicase activity of Sgs1 was not required[20]. In line with this, a largely Sgs1 helicase activity-independent role was shown for the STR complex in D-loop processing in vivo[15]. This indicates that the activity of the yeast TR complex has a dominant contribution to D-loop processing, and Sgs1 has more of a structural role. However, we can envision that Sgs1, and also BLM, may aid the process by generating short ssDNA segments on which the TR or TRR complex can act more efficiently.

In summary, the above results, together with genetic evidence from various organisms, highlight the involvement of RecQ helicase-topoisomerase complexes in D-loop disruption and HR regulation[15,31]. However, it is still not fully understood how BLM is involved in these processes at the precise mechanistic level and whether it constitutively works in complex with TRR. A large body of evidence establishes the essential role of BTRR or STR complexes during meiosis, somatic DNA repair, Holliday junction dissolution and/or resolution of late-replication intermediates, and previous results also support the function of STR in D-loop disruption, highlighting a strong interdependence of these proteins[15,20,50,55,56]. Importantly, however, context-dependent, extensive but only partial colocalization of BLM and topoisomerase IIIα has been reported in mitotic cells[57] and only partial colocalization of the two proteins was observed on meiotic chromosomes[58], raising the idea that BLM may in some contexts function independently of the TRR complex. Based on our findings and those of others mentioned above, we can envision differential roles for the BTRR complex and BLM alone in the regulation of HR-related processes (Fig. 10).

When BLM is in complex with TRR, the efficient D-loop disruption activity of the complex may lead to elimination of nascent strand invasions, brought about probably mainly by the topoisomerase activity as proposed for the STR complex[15,20]. In somatic cells, this activity may serve quality-control functions to (i) increase the fidelity of homology search, (ii) inhibit nonallelic

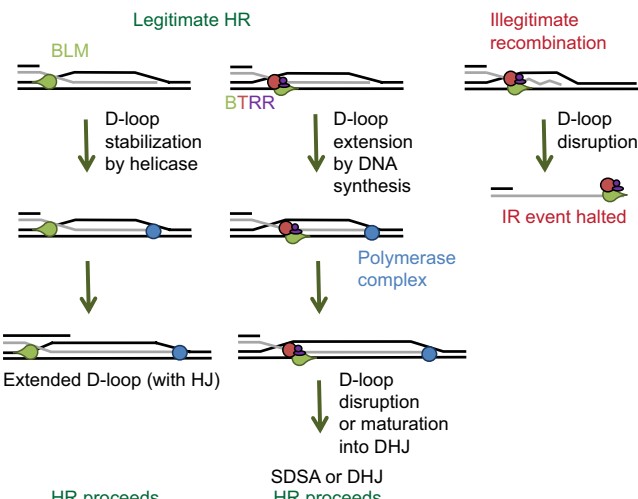

**Fig. 10 BLM can either disrupt or stabilize D-loops, depending on the presence of TRR.** Due to its balanced D-loop binding orientations, BLM (green droplet) alone (in the absence of TRR) (red and purple circles) may load onto the D-loop structure in a way that leads to stabilization of the D-loop upon subsequent dsDNA unwinding (left pathway). Extensive unwinding may lead to (probably infrequent) conversion of the D-loop into a single Holliday junction. The formed structure will be processed further based on the cellular context, e.g., by DNA polymerase (blue circle)-mediated extension of the strand invasion. In contrast, the BTRR complex may efficiently disrupt nascent D-loops, and structures escaping disruption can be extended by DNA polymerases (middle pathway). In principle, D-loops that mature via polymerase extension can also be disassembled by the BTRR complex to facilitate the SDSA pathway, whereas in a situation where BLM would act alone, it could potentially stabilize these structures as described for nascent D-loops (left pathway). Alternatively, HR may proceed through the formation of a DHJ structure via second-end capture. Disruption of nascent illegitimate D-loops may lead to additional rounds of strand invasions and may promote quality control of homologous recombination processes (right pathway).

recombination, (iii) suppress recombination between homologous chromosomes during somatic repair or between sister chromosomes during meiosis, and (iv) prevent the formation of multi-chromosomal joints. Recently, the *C. elegans* Him-6 helicase, an ortholog of BLM, was found to be involved in selective removal of illegitimate recombination events[17]. We proposed a model of how *E. coli* RecQ helicase may be able to selectively disrupt illegitimate D-loop structures based on its geometry- and DNA sequence-dependent unwinding activities[19,59], though additional experiments are required to determine the extent to which BLM and related eukaryotic helicases share the specific unwinding activities of RecQ that result in selective disruption of illegitimate D-loop structures.

In addition to quality-control functions, BTRR-mediated D-loop disruption may be involved in the tight regulation of crossover (CO) frequency during meiosis. BLM, Sgs1, and other RecQ-family helicases together with topoisomerases in multiple species were shown to be important during meiosis to ensure progression of HR into CO formation at designated sites and to inhibit formation of COs at nondesignated loci[54,55,60–63] (reviewed in[4,52]). In line with this, BLM localizes to multiple foci together with RAD51 and DMC1 recombinases on meiotic chromosomes during late leptotene through pachytene meiotic phases, and the number of BLM foci exceeds the number of expected COs[64]. Similarly, in *C. elegans*, Him-6 is present both at presumed NCO (noncrossover) and CO sites until late pachytene, and is required for stabilization of COs[61,65]. The above-

mentioned observations may underpin a mechanism in which helicase–topoisomerase complexes dynamically disrupt nascent and/or extended strand invasions to regulate recombination outcome, in addition to the NCO-forming dHJ dissolution pathway, and only a small fraction of invasions that are stabilized by procrossover proteins can escape disruption[4,15,31,63,66]. Whereas multiple protein factors were shown to be important in stabilization (reviewed in[67]), the role of RecQ-family helicases in this step has not been clarified. Based on the partial colocalization of BLM and topoisomerase IIIα[58] and BLM's observed balanced D-loop binding orientations, we can hypothesize a distinct function for BLM alone and propose two scenarios. If BLM binds in an orientation that leads to strand-invasion disruption in our assay, D-loop disruption will probably be inhibited due to the presence of active recombinases, until assembly of the BTRR complex or inactivation of the recombinase nucleoprotein filament. Importantly, however, a BLM binding orientation that leads to unwinding of the dsDNA region running "outward" from the junction would lead to stabilization of the D-loop. This activity, in principle, could lead to the subsequent formation of a HJ structure (Fig. 10), which could be extended by polymerase activity, disrupted in the following steps, or serve as a substrate for HJ-specific endonuclease (resolvase) enzymes (HJ resolvases are reviewed in ref. [68]). In addition to the altered D-loop processing dynamics, these structures may be further stabilized by procrossover factors to ensure progression into CO-forming pathways. Similar scenarios may occur in somatic cells as well, possibly with different additional steps channeling the reaction toward NCO outcomes.

D-loops that escape initial disruption during somatic repair or meiosis and mature via the DNA-strand extension activity of DNA polymerase can be also disassembled to channel HR into the SDSA pathway. As SDSA leads exclusively to NCO events, it may promote error-free DNA repair in somatic cells and has also been proposed to be one of the main mechanisms governing HR into NCO outcomes at non-CO-designated loci during meiosis. Alternatively, D-loops can be converted to DHJs via capture of an available second broken DNA end[2]. The DHJ can be processed by resolvase complexes (some of which involve BLM)[69] to lead to NCO or CO outcomes or dissolved by the BTRR complex, leading to NCO products, depending on the cellular context[8,25]. Whereas *Drosophila melanogaster* BLM[21] and yeast Sgs1 were implicated in promotion of the SDSA pathway[63,70], Sgs1 was shown not to influence the efficiency of D-loop extension significantly[15] and the absence of BLM in cell lines did not decrease the frequency of SDSA[71]. These results indicate that the role of these helicases in disruption of extended D-loops may be species-specific and the role of BLM and/or BTRR (and STR) complex may be limited to the processing of nascent invasions. These processes may further be regulated by additional, yet unexplored factors. Further research is warranted to clarify the role of these enzymes in disruption of extended D-loops and to test whether the activity that we propose could manifest during such processes.

Our currently proposed activity of BLM in HR highlights the fact that BLM interaction partners can markedly influence the outcomes of BLM-associated D-loop processing. In addition to D-loop disruption, the stabilization of D-loops by helicases, a rarely investigated activity, should also be considered in biochemical and cellular models. Further studies will be required to clarify how other factors, including additional properties of the D-loop structure, post-translational modifications and various proteins present at D-loops, influence D-loop processing.

## Methods

**Reagents.** All reagents were from Sigma-Aldrich unless otherwise stated. ATP was from Roche Applied Science. For concentration determination, the $\varepsilon_{260}$ value of

$10300$ M$^{-1}$ cm$^{-1}$ nt$^{-1}$ was used for nonhomopolymeric oligonucleotides. DNA concentrations are expressed as those of oligo- or polynucleotide molecules (as opposed to those of constituent nt units), unless otherwise stated.

**Cloning, protein expression, and purification**. The cloning of human BLM$^{FL}$ (aa 1–1417) and BLM$^{CR}$ (aa 642–1290) is described in refs. [47,72,73]. The coding region for BLM-dH (aa 642–1191) was amplified by PCR using the pTXB3/BLM$^{CR}$ plasmid (encoding BLM aa. 642–1290)[73] as template, and subcloned between the NcoI and SapI sites of pTXB3. Cloning of BLM-dWH (aa 642–1077) is described in ref. [32]. Cloning of RecQ constructs is described in refs. [19,74,75]. All constructs were verified by DNA sequencing.

BLM$^{FL}$, BLM$^{CR}$, BLM-dH, and BLM-dWH were expressed and purified based on refs. [32,47,72,73]. RecQ proteins were purified based on ref. [19]. TRR expression and purification was performed according to ref. [76]. Expression and purification procedures are briefly described below.

**Human BLM$^{FL}$**. A pYES2 plasmid containing the coding sequence of the full-length human BLM protein and a C-terminal His tag was electroporated into JEL1 yeast cells. Transformed cells were incubated in selective media (containing 3 v/v% glycerol and 2 v/v% dl-lactate and no sugar) at 30 °C. Expression was started with 2 w/v% galactose, and cultures were incubated at 20 °C for 24 h with shaking. Cells were harvested by centrifugation and equal volume of lysis buffer (50 mM sodium phosphate, pH 7.0, 500 mM KCl, and 10 v/v% glycerol, EDTA-free protease-inhibitor mixture (Roche)) was added. The mixture was immediately frozen in liquid N$_2$ in ~5-ml bullets. Bullets were quickly ground in a coffee grinder to lyse the cells and the sample was melted in 15-ml tubes in a water bath at room temperature. The following purification steps were carried out at 4 °C. The lysate was cleared by centrifugation (30.000xg, Beckman JA-20 rotor, 30 min) and the supernatant was loaded onto a Ni-NTA column (10 ml, QIAGEN) equilibrated with a buffer A (50 mM Na phosphate, pH 7.0, 500 mM NaCl) plus 15 mM imidazole. The column was washed with buffer A plus 50 mM imidazole. Protein was eluted with a linear imidazole gradient (50 mM–1.5 M in buffer A, 50 ml, 1 ml/min). Fractions containing BLM were checked with SDS-PAGE. Purest fractions were pooled and dialyzed against Storage Buffer (100 mM Tris-HCl, pH 7.5, 500 mM NaCl, 1 mM DTT, and 10 v/v% glycerol) overnight.

**Human BLM$^{CR}$, BLM-dH, and BLM-dWH**. pTXB3 plasmids containing the coding sequence for the given BLM helicase construct in-frame with downstream intein- and chitin-binding domain tags were transformed into Rosetta (DE3) E. coli cells (New England Biolabs). Cells were grown in 2YT media supplemented with ampicillin and chloramphenicol. Protein expression was induced with 0.2 mM IPTG (Sigma) at OD$_{600}$ = 0.6 and cells were incubated for 16 h at 18 °C with shaking. Cells were harvested with centrifugation. Buffer CH (50 mM Tris-HCl pH 8.0, 0.5 M NaCl, 1 mM EDTA, 10 v/v% glycerol, and 0.1 v/v% Triton X-100) was added to the pellet (40 ml/l culture) and after dounce homogenization cells were sonicated on ice. The following purification steps were carried out at 4 °C. The lysate was cleared by centrifugation (30.000xg, Beckman JA-20 rotor, 30 min) and the supernatant was loaded onto an 8-ml chitin resin (New England Biolabs) equilibrated with CH buffer. After loading, the column was washed with CH buffer and CH buffer supplemented with 50 mM DTT. After overnight incubation, the tag-free protein was eluted with CH buffer. Pooled fractions were diluted 2.5 times with 50 mM Tris-HCl pH 8.0 and loaded onto a 5-ml HiTrap Heparin HP column (Cytiva) equilibrated with buffer HPA (50 mM Tris-HCl, pH 8, 0.1 mM EDTA, 10 v/v% glycerol, and 1 mM DTT) plus 200 mM NaCl. Protein was eluted with a linear salt gradient (200 mM–1 M NaCl in buffer HPA, 50 ml, 1 ml/min). Fractions containing the given protein were pooled and diluted three times with 20 mM Tris-HCl pH 8, and loaded onto a CM sepharose FF column (GE Healthcare) equilibrated with CM buffer (20 mM Tris-HCl pH 8, 0.1 mM EDTA, 10 v/v% glycerol, and 1 mM DTT) plus 150 mM NaCl. Protein was eluted with linear salt gradient (150 mM–1 M NaCl in CM buffer, 20 ml, 1 ml/min). Fractions containing the protein were pooled and dialyzed against a storage buffer (50 mM Tris-HCl pH 7.5, 200 mM NaCl, 10 v/v% glycerol, and 1 mM DTT) overnight.

***E. coli* RecQ, RecQ\*, RecQ-dH**. pTXB3 plasmids containing the coding sequence for the given RecQ helicase construct in-frame with downstream intein- and chitin-binding domain tags were transformed into E. coli B ER2566 cells (New England Biolabs). Cells were grown in 2YT media supplemented with ampicillin. Protein expression was induced with 0.2 mM IPTG (Sigma) at OD$_{600}$ = 0.6 and cells were incubated for 16 h at 18 °C with shaking. Cell lysis and purification on chitin column were done as described for truncated BLM constructs. RecQ and RecQ\* were further purified on heparin column as described for truncated BLM constructs; the CM column was omitted.

RecQ-dH binds weakly to the heparin column. Therefore, instead of the heparin column, this protein was purified on a 1-ml Mono Q column (Mono Q 5/50 GL, Cytiva) as follows. The eluted fractions from the chitin column were dialyses in MQ buffer (50 mM Tris-HCl, pH 8.5, 0.1 mM EDTA, 10 v/v% glycerol, and 1 mM DTT) plus 20 mM NaCl overnight. The sample was loaded onto the Mono Q column equilibrated with MQ buffer plus 20 mM NaCl. Protein was eluted with a linear NaCl gradient (20 mM–1 M Nacl in MQ buffer, 50 min, 1 ml/min).

Fractions containing the purified protein were pooled and dialyzed against a storage buffer (50 mM Tris-HCl, pH 7.5, 200 mM NaCl, 10 v/v% glycerol, and 1 mM DTT) overnight.

**Human TRR**. For expression of the human TRR complex, Rosetta (DE3) E. coli cells were co-transformed with pCDFDuet (encoding RMI1 and RMI2) and pET29H2 plasmids (encoding his-tagged TOPOIIIα) provided by Dr. Ian D. Hickson from the University of Copenhagen[76]. Briefly, cells were grown at 37 °C in LB media containing 10-10 µg/ml of streptomycin, kanamycin and chloramphenicol. At OD$_{600}$ = 0.6, expression was induced with 0.3 mM IPTG for 3 h at 25 °C. Cells were harvested and washed with Lysis Buffer (50 mM Tris-HCl pH 7.5, 0.5 M NaCl, 10 v/v% glycerol, 0.1 v/v% IGEPAL, 1 mM DTT, 40 mM imidazole, 1 mM PMSF, and protease-inhibitor tablet) (Pierce Thermo Fisher) before dounce homogenization and sonication. After the removal of cell debris by centrifugation, the lysate was affinity-purified on a 10-ml HisPur Ni-NTA (Thermo Fisher) column equilibrated with Wash Buffer (same as Lysis Buffer without PMSF and protease-inhibitor tablet). The complex was eluted with Wash Buffer supplemented with 500 mM imidazole and further purified on a 5-ml HiTrap Heparin HP column (Cytiva) in HP buffer (50 mM Tris-HCl pH 7.5, 10 v/v% glycerol, 0.1 mM EDTA, and 1 mM DTT) with a linear gradient of 200 mM to 1 M NaCl. Fractions containing all three proteins were pooled together and the eluate was dialyzed overnight at 4 °C in Storage Buffer (50 mM Tris-HCl pH 7.5, 10 v/v% glycerol, 200 mM NaCl, and 1 mM DTT).

Protein purity was checked by SDS-PAGE (Supplementary Fig. 1). Concentrations of purified proteins were measured using the Bradford method, the concentration of the TRR complex was calculated based on the combined molecular weight of the three proteins. Purified proteins were flash-frozen and stored in liquid N$_2$ in 20-µL droplets.

**Sequences of DNA substrates (5′–3′)**. D1: GACGCTGCCGAATTCTACCAGT GCCTTGCTAGGACATCTTTGCCCACCTGCAGGTTCACCC

D2: GGGTGAACCTGCAGGTGGGCGGCTGCTCATCGTAGGTTAGTTG GTAGAATTCGGCAGCGTC

D3: Cy3-TAAGAGCAAGATGTTCTATAAAAGATGTCCTAGCAAGGCAC

D4: TATAGAACATCTTGCTCTTA

ss54-FLU: TCCTTTTGATAAGAGGTCATTTTTGCGGATGGCTTAGAGCT TAATTGCGCAACG -fluorescein

I-trap (DL4 trap strand): same as D3 but without Cy3 label

R-trap: AATTCGGCAGCGTC-fluorescein

L-trap: Fluorescein-GGGTGAACCTGCAGGT

3 T: D3 + D4

DL3: D1 + D2 + D3

DL4: D1 + D2 + D3 + D4

**DNA-substrate preparation**. DNA substrates were prepared as described in Harami et al.[19]. Equimolar amounts of the applicable oligonucleotides were mixed in a buffer comprising 10 mM Tris-HCl, pH 7.5, and 50 mM NaCl. Samples were boiled and were left to cool down to room temperature overnight. Samples were purified on a Mono Q anion-exchange column (Mono Q 5/50 GL, Cytiva) using a 0.01–1 M NaCl gradient for elution and fractions were analyzed by PAGE. Fractions containing the desired DNA structures were desalted by using an Amicon Ultra centrifuge filter (Millipore). DNA substrates were aliquoted and stored at –80 °C. Oligonucleotide sequences are listed above.

**DNA unwinding kinetic experiments**. DNA unwinding experiments using the I-trap alone (unlabeled version of the invading ssDNA, see above for oligonucleotide sequences) were performed as described in Harami et al.[19]. During the experiment, preformed complexes of the given helicase construct (100 nM except for BLM-dWH (1 µM), final reaction concentrations stated) and the given Cy3-labeled DNA substrate (DL4: four-stranded D-loop-like substrate, DL3: three-stranded D-loop-like substrate, and 3T: two-stranded, 3′-tailed DNA structure; Fig. 2b) (30 nM) were mixed with excess ATP (3 mM) and a large amount (3 µM) of ssDNA trap strand ("I-trap", with a sequence identical to that of the labeled invading strand) in order to inhibit DNA reannealing and enzyme rebinding to labeled substrates after dissociation. Reactions (performed at 37 °C) were stopped by the addition of 8 mM EDTA and 0.16 w/v% SDS at the given timepoints. In control reactions (zero time points) the helicase. DNA complex was first mixed with the stop solution and then ATP and DNA trap strands were added. These control reactions confirmed that the stop solution completely inhibits helicase activity. D-loop disruption experiments involving the L-trap or R-trap were performed as the I-trap-only experiments, except that, in addition to the unlabeled DL4 trap strand (I-trap, 1.5 µM), fluorescein-labeled ssDNA trap strands L-trap or R-trap (Fig. 4) were also added to 1.5 µM final concentration.

DNA species were separated on 12 w/v% native polyacrylamide gels via electrophoresis (PAGE) (Fig. 2b and c, Supplementary Figs. 4–6,8,11–12 and 14). Gels were scanned for the Cy3 signal present on the invading strand and in L- and R-trap experiments also for the fluorescein signal present on L- or R-traps. DNA bands were assigned to DNA species based on control measurements (Fig. 2b for I-trap; Supplementary Fig. 6 for L- and R-trap experiments) and the fractions of

DNA species were determined by fluorescence densitometry (Figs. 4a, 6–8, Supplementary Figs. 2b–c,7–10 and 13–14). For gels scanned for Cy3 signal, the ssDNA band (D3 oligonucleotide, see above for oligonucleotide sequences); for gels scanned for fluorescein signal, the L- trap or R-trap oligonucleotide bands are present as molecular weight markers.

Helicase-mediated D-loop disruption assays with TRR were performed in the presence of 1.2 μM TRR if not otherwise indicated. Before addition of the TRR complex to D-loop experiments, to avoid changes in the assay buffer conditions, the storage buffer of TRR was exchanged to the assay buffer and the concentration of TRR was remeasured.

**Global kinetic modeling of D-loop processing experiments.** Global fitting kinetic analysis was performed using KinTek Global Kinetic Explorer 4.0[77,78], based on mass action rate equations accounting for all steps depicted in Fig. 3 (I-trap only) or Supplementary Fig. 7 (L- and R-trap experiments), and the initial fractions of the DNA species shown at zero time in Fig. 4a or Figs. 6–8; Supplementary Fig. 13. Fitting of I-trap data was done identically as for RecQ in our previous work[19].

For robust fitting, the extended model was globally fitted to the L-trap or R-trap DL4 data obtained using the Cy3 and fluorescein signals. During fitting, the DLR pathway was fixed to zero, as the five-stranded DL5R (Fig. 5b) structure was not observed for any helicase construct (Supplementary Figs. 5,12b). To further increase the robustness of the fit, we included the results of the 3T and DL3 unwinding experiments, obtained only in the presence of I-trap and the absence of TRR, for BLM (Fig. 4a or Fig. 8 and Supplementary Fig. 13 data obtained in the absence of TRR) and RecQ (data from ref. [19]) constructs in the fitting process. The usage of 3T results for this purpose is appropriate as 3T unwinding kinetics are independent of the presence of L-trap or R-trap (neither of these trap strands share sequence homology with any part of 3 T). For fits to measurements performed with the TRR, 3T and DL3 data obtained in the presence of TRR were used (Fig. 8 and Supplementary Fig. 13 data with TRR). Accordingly, in control experiments, neither L-trap nor R-trap interacted with 3 T or with the invading ssDNA strand that is part of 3 T (Supplementary Fig. 6). DL3 data obtained with I-trap only can also be independently used in modeling, as the presence of L-trap or R-trap, while it makes the DLL (and, in principle, DLR) pathways detectable, it does not influence the partitioning of other pathways. The practically identical ssDNA accumulation kinetics observed in L-trap, R-trap, and I-trap-only DL4 experiments supports this assumption (Supplementary Fig. 8b).

For parsimony, successful unwinding of any of the dsDNA segments of the DNA substrates was modeled to occur at a single rate constant $k_U$ in both models, as the lengths of these segments were similar (21-bp invasion, 20-bp other segments). Unwinding ($k_U$) and rebinding rates ($k_r$) were initialized manually. Pathway fractionation for BLM I-trap experiments was initialized based on the initial fractional change of DL4 and 3T DNA species. For L- and R-trap experiments (Figs. 6 and 7), pathway fractionation was initialized based on previous I-trap results and the initial fractional change of the DL5L DNA species. For TRR and related control experiments (Fig. 8, Supplementary Fig. S13), pathway fractionation was initialized based on the results of Figs. 6 and 7. In cases when fits did not converge in the initial set, parameters were manually readjusted, and fits were refined. For fitting, means and SD data of the DNA species fractions at individual time points were used as input.

Numerical results of global fits are reported in Supplementary Tables 1–3. For Supplementary Tables 2–3, fitting robustness/uncertainty for each parameter was separately estimated as the SD of 5 fitting runs starting from randomly initialized pathway fractionations without manually readjusting the fitting results. We note that the best-fit values for minor fractions of DNA-binding configurations and for pathways occurring after enzyme rebinding to various products and intermediates from the trap strand after single-round unwinding events (pathways PI, PE, and PN in Supplementary Fig. 7) have high uncertainties. This feature reflects that these parameters have an inherently low effect on most of the experimental readouts. Nevertheless, the precise value of these parameters is marginal to the conclusions drawn from the modeling results, which are based on the robustly determined fractions of major DNA binding configurations and DNA-processing pathways.

**Fluorescence-anisotropy titrations.** These experiments were performed as described in Harami et al.[19]. Measurements were done in a buffer containing 50 mM Tris-HCl pH 7.5, 50 mM NaCl, 1 mM DTT, 5 mM MgCl₂, and 50 μg/mL BSA. In all, 10 nM ss54-FLU or fluorescein-labeled DL4, DL3 (for these structures, a fluorescein-labeled version of D3 strand was used; see oligonucleotide sequences above) was titrated with increasing concentrations of proteins at 25 °C. In case of competitive titration experiments, Cy3-labeled DL4, DL3, or 3 T substrates were used to compete with ss54-FLU binding to the given helicase construct, whereas ss54-FLU signal was monitored at 25 °C. For competitive titrations, protein and DNA concentrations are shown in the figures. Fluorescence anisotropy was measured in a Synergy H4 Hybrid Multi-Mode Microplate Reader (BioTek).

**Data analysis.** Means ± SEM values are reported in the paper, unless otherwise specified. Sample sizes (n) are given for number of ensemble in vitro measurements performed using independent protein preparations (biological replicates, n = 3,

unless otherwise specified). Data analysis was performed using OriginLab 8.0 (Microcal Corp.). Pixel densitometry was performed using the GelQuant Pro v12 software (DNR Bio Imaging Ltd.).

**Reporting summary**. Further information on research design is available in the Nature Research Reporting Summary linked to this article.

## Data availability
The data that support this study are available from the corresponding authors upon reasonable request. The pdb file 4O3M used for the presentation of the BLM 3D structure was obtained from the Protein Data Bank. Source data are provided with this paper.

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

## Acknowledgements

This work was supported by the Human Frontier Science Program (RGY0072/2010 to M.K. and K.C.N.), the "Momentum" Program of the Hungarian Academy of Sciences (LP2011-006/2011 to M.K.), ELTE KMOP-4.2.1/B-10-2011-0002, NKFIH K-116072, NKFIH K-123989, and NKFIH ERC_HU 117680 grants to M.K. This work was supported in part by the Intramural Research Program of the National Heart, Lung, and Blood Institute, National Institutes of Health (HL001056-07 to K.C.N.). M. G. was supported by the Marie Sklodowska–Curie Fellowship Programme (H2020-MSCA-IF-2014-657076 to M. G.). G.M.H. was supported by the Premium Postdoctoral Program of

the Hungarian Academy of Sciences (PREMIUM-2017-17 to G.M.H.). Z.J.K. and J.P. are supported by the New National Excellence Program of the Ministry for Innovation and Technology (Grants ÚNKP-21-3 to Z.J.K. and ÚNKP-19-2 to J.P.) and Z.J.K. is supported by the Co-operative Doctoral Program of the Ministry of Innovation and Technology financed from the National Research, Development and Innovation Fund. This work was completed in the ELTE "SzintPlusz" Thematic Excellence Programme supported by the Hungarian Ministry for Innovation and Technology. We are grateful to Drs. Ian Hickson and Kata Sarlós for providing the expression vectors for the TRR complex.

## Author contributions

G.M.H., K.C.N., and M.K. designed research. G.M.H., J.P., Z.J.K., M.G., and H.H.-P. performed experiments. G.M.H., J.P., Y.S, K.C.N., and M.K. analyzed data. G.M.H and M.K. wrote the paper.

## Funding

## Competing interests

The authors declare no competing interests.
