## [Peer Review File · Nature Communications]

Reviewers' comments:

Reviewer #1 (Remarks to the Author):

The manuscript by Harami et al., focuses on understanding the molecular mechanism by which BLM can disrupt or stabilize D- loops. The molecular mechanism to resolve such D-structures by BLM helicase remains elusive and the current models of the different pathways are presented in Fig. 1A. This is essential to understand as many pathophysiological are associated with genome instability and understanding BLM action of mechanism is likely to shed a light on ageing, cancer and other related human diseases.

The author presented, based on their previous studies as well, that they have "devise an analytical method to resolve all possible pathways of D-loop processing". With this regard it would be more appropriate for the authors to moderate 'all possible' to 'all known pathways...'.
The author major argument is that BLM acts as a quality control regulator by impact the intrinsic balance between the D-loop disruption and its stabilization. This is decisive for the pathway that follows after. However, what are the effectors or the molecular mechanism/s which balance this check point of HR pathways selection are not provided.

The experimental data presented in this work is well quantified, and with high quality standards, and presented with excellent clarity and flow. The statistical analysis is satisfactory and performed in all of the measurements. On that note, it should be clarifying why in some figures 'SEM' was used in the legends and then n=3 rather than reporting SE as explained in the method section.

The novelty of the paper is questionable since all the experimental set up were previously presented in an earlier publication with the exception of Figure 4 and are performed here in the context of BLM helicase including the similar protein constructs.

The key experiment would have to reveal the molecular basis which differ RecQ from BLM helicase either by finding additional accessory proteins or the molecular mechanism itself that is affecting this bias activity. This could greatly benefit the work to distinguish it from the previous publication of the same group in PNAS. One approach would be to construct chimeras between RecQ and BLM helicases that will exhibits a switch in the enzyme properties and by that identifying the molecular basis for the discrepancies in processing D -loop structure between the two enzymes.

The fact that a helicase can serve as a homologues recombination quality control machinery for 'D-loop' processing and serve as a sensor beyond being helicases per se is very interesting. This is likely to be very revealing for the community of helicase with limitation beyond this field since such quality control sensing mechanism have been proposed before for other systems such as NMD.

Specific comments

1. The global fitting is a very important contribution to the manuscript. However, some of the fitted parameters are just within the same magnitude of the error of the fitted parameter. First, the authors should provide an explanation for that. Second, some of the species fractions formed in the tables are almost negligible. It may be also wise to present the data in the table graphically.
2. The section "The accessory domains of BLM maintain balance between D-loop disruption and extension" seems quite long and the authors should consider to split it up into to two section in order to enhance the focus of this long section.
3. It is not clear what is the reference for the statement "not to differences in enzyme processivities" is the processivities (Figure 7a) relays on previous determination, or it is based on this presented data and if so how was this determined. Is this conclusion is only based on Fig 7?
4. The authors should reference or comment on how reproducible the analysis of the fluorescence gels is in terms of the dynamic range of the fluorophore they have used versus radioactive probes.
5. "The WHD promotes processive unwinding whereas the HRDC domain mediates pausing and shuttling on DNA hairpin substrate" is not in line with the fact that there is no evidence for BLM being processive and it is also contradictory to the discussion (Page 15 middle paragraph). The authors should better define this presumably activity of processiveness by BLM syndrome.

6. The work by Nimonkar et al. seems to be in line with the idea that accessory proteins are critical for the end product resection, hence it would have been essential to test with these accessory proteins utilizing this and previous work very clever and intelligent substrates the processing of the D-loop.

7. The manuscript is very well written, there is one tiny spelling mistake:
Page 16 of the manuscript, last paragraph, line 2, "DNA strand" is misspelt as "DNA stand".

Reviewer #2 (Remarks to the Author):

Harami et al.

Intrinsic balance between D-loop disruption and stabilization by human Bloom's syndrome helicase"

BLM helicase and its homologs in model systems such as Sgs1 in the budding yeast play multiple roles during homologous recombination. Both are involved in the initial resection of the double-strand break in conjunction with the DNA2 nuclease activity. BLM has also been shown to destabilize RAD51-ssDNA filaments under certain biochemical conditions, and the physiological relevance of this activity remains unproven. Moreover, BLM and Sgs1 have been shown to dissociate protein-free D-loop, the product of homology search and DNA strand invasion. Furthermore, from genetic observation during meiotic recombination in budding yeast, it was inferred that Sgs1 stabilizes the crossover dedicated Single-End Invasion intermediate. Finally, BLM and its complex partners TOPOIIIalpha-RMI1-RMI2, as the human yeast homologous complex Sgs1-Top3-Rmi1, can process double Holliday junction to non-crossover products in a reaction that requires both the helicase and topoisomerase (decatenase) activity of the complex.

The present manuscript utilizes purified BLM wild type and mutant constructs on short, oligonucleotide-based substrates to dissect the activities of BLM on D-loops. The authors conclude from the in vitro observations that BLM balance D-loop formation and disruption in vivo.

The fundamental problem is that BLM does not work in isolation. First, it functions in complex with TOPOIIIalpha-RMI1-RMI2. Second, the enzyme functions in the context of the HR machinery RAD51-RAD54 and other proteins which decorate the DNA intermediates during HR. In fact, for the yeast Sgs1 this has been elaborated, and it is disappointing that this relevant work was not considered. While Sgs1 dissociates protein-free D-loops like BLM, it cannot dissociate D-loops in reconstituted reactions with Rad51, Rad54, RPA in isolation. Rather under reconstituted conditions, the topoisomerase activity of Top3-Rmi1 or the STR complex actively dissociates D-loops (Fasching et al. 2015 Mol. Cell). These in vitro observations are congruent with in vivo data showing that D-loop intermediates accumulate in Sgs1-deficient cells, but this accumulation is independent of its ATPase (i.e. helicase activity) and only dependent on the Top3 decatenase activity (Piazza et al. 2017 Cell 2017, Piazza et al. 2019 Mol. Cell). While the authors selectively discuss work with yeast Sgs1 to support their conclusions, they do not discuss the relevant work done in vivo and in vitro with the yeast enzymes cited above.

The present conclusions severely muddle the field and the paper should not be published. The biochemical characterization of BLM on protein-free substrates adds biochemical information, but the results will likely bear little biological significance. Hence submission to a more specialized journal deems more appropriate, which would still require significant rewriting of conclusions and interpretations as well as discussions of the prior work in yeast.

Response to reviewers

Reviewer #1 (Remarks to the Author):

“The manuscript by Harami et al., focuses on understanding the molecular mechanism by which BLM can disrupt or stabilize D- loops. The molecular mechanism to resolve such D-structures by BLM helicase remains elusive and the current models of the different pathways are presented in Fig. 1A. This is essential to understand as many pathophysiological are associated with genome instability and understanding BLM action of mechanism is likely to shed a light on ageing, cancer and other related human diseases.

The author presented, based on their previous studies as well, that they have “devise an analytical method to resolve all possible pathways of D-loop processing”. With this regard it would be more appropriate for the authors to moderate ‘all possible’ to ‘all known pathways...’.”

We changed the text to read “all conceivable D-loop unwinding orientations” as those in the model were defined on a logical basis considering BLM’s known DNA binding and unwinding properties.

“The author major argument is that BLM acts as a quality control regulator by impact the intrinsic balance between the D-loop disruption and its stabilization. This is decisive for the pathway that follows after. However, what are the effectors or the molecular mechanism/s which balance this check point of HR pathways selection are not provided.

The experimental data presented in this work is well quantified, and with high quality standards, and presented with excellent clarity and flow. The statistical analysis is satisfactory and performed in all of the measurements. On that note, it should be clarifying why in some figures ‘SEM’ was used in the legends and then n=3 rather than reporting SE as explained in the method section.”

We thank the Reviewer for acknowledging the quality of the data and their presentation. In the new version we consistently use ‘SEM’ to signify the standard error of the mean.

“The novelty of the paper is questionable since all the experimental set up were previously presented in an earlier publication with the exception of Figure 4 and are performed here in the context of BLM helicase including the similar protein constructs.

The key experiment would have to reveal the molecular basis which differ RecQ from BLM helicase either by finding additional accessory proteins or the molecular mechanism itself that is affecting this bias activity. This could greatly benefit the work to distinguish it from the previous publication of the same group in PNAS. One approach would be to construct chimeras between RecQ and BLM helicases that will exhibits a switch in the enzyme properties and by that identifying the molecular basis for the discrepancies in processing D -loop structure between the two enzymes.”

The previous version of the manuscript defined a novel D-loop processing mechanism for human BLM helicase which had never been observed previously, and which markedly differs from the behavior we published earlier for *E. coli* RecQ (Harami et al. 2017 PNAS). Nonetheless, the addition of the new data in the revised version on the striking regulatory effect of TRR on BLM’s mechanism crucially increased the level of insight into this central HR process.

“The fact that a helicase can serve as a homologues recombination quality control machinery for ‘D-loop’ processing and serve as a sensor beyond being helicases per se is very interesting. This is likely to be very revealing for the community of helicase with limitation beyond this field since such quality control sensing mechanism have been proposed before for other systems such as NMD.”

We thank the Reviewer for acknowledging the importance of the insights obtained, which is greatly enhanced by the new data.

Specific comments:

1. *The global fitting is a very important contribution to the manuscript. However, some of the fitted parameters are just within the same magnitude of the error of the fitted parameter. First, the authors should provide an explanation for that. Second, some of the species fractions formed in the tables are almost negligible. It may be also wise to present the data in the table graphically.*

We have inserted a new paragraph into Materials and Methods (p. 11) to note that the best-fit values for minor fractions of DNA binding configurations and for pathways occurring after enzyme rebinding to various products and intermediates from the trap strand after single-round unwinding events (pathways PI, PE and PN in Supplementary Fig. 7) have high uncertainties. This feature reflects that these parameters have an inherently low effect on most of the experimental readouts. Nevertheless, the precise value of these parameters is marginal to the conclusions drawn from the modeling results, which are based on the robustly determined fractions of major DNA binding configurations and DNA processing pathways.

We agree that some of the fractions are negligible for some constructs, and these values often have high uncertainties. These steps/pathways could be omitted, without much effect, from global fitting of data for the given construct; however, we did not want to alter the number of floating parameters in the model for the different constructs. Steps in the model, which involve rebinding of the enzyme from the trap strand after single-round unwinding event and which have no directly detectable product DNA structures have also a high uncertainty. Importantly, however, while these steps are important to explain the disappearance of some intermediate DNA structures (such as DL5L), they do not influence significantly the initial partitioning of D-loop binding/processing orientations determined by modeling.

Following the Reviewer's suggestion, to facilitate visual comparison between numerical values, the fractions for various pathways for various proteins are graphically shown in Figs. 3b, 6d and 7. We have also inserted a new figure (Fig. 7) to graphically show the main findings of the modeling in terms of D-loop processing directions.

"2. The section "The accessory domains of BLM maintain balance between D-loop disruption and extension" seems quite long and the authors should consider to split it up into two section in order to enhance the focus of this long section."

This section has been modified and shortened in the new version.

"3. It is not clear what is the reference for the statement "not to differences in enzyme processivities" is the processivities (Figure 7a) relies on previous determination, or it is based on this presented data and if so how was this determined. Is this conclusion is only based on Fig 7?"

This section has been modified in the new version, and the statement in question has been removed.

"4. The authors should reference or comment on how reproducible the analysis of the fluorescence gels is in terms of the dynamic range of the fluorophore they have used versus radioactive probes."

Following the Reviewer's suggestion, we have inserted a new figure (Supplementary Fig. 2a in the new version) showing the dynamic range of the used fluorescent detection method. Practically, we can reliably detect amounts as low as 10 fmol fluorophore and the signal is linearly dependent on the concentration in the range relevant to our assay.

"5. "The WHD promotes processive unwinding whereas the HRDC domain mediates pausing and shuttling on DNA hairpin substrate" is not in line with the fact that there is no evidence for Blm being processive and it is also contradictory to the discussion (Page 15 middle paragraph). The authors should better define this presumably activity of processiveness by BLM syndrome."

This part has been removed, and the new version focuses exclusively on D-loop processing.

"6. The work by Nimonkar et al. seems to be in line with the idea that accessory proteins are critical for the end

product resection, hence it would have been essential to test with these accessory proteins utilizing this and previous work very clever and intelligent substrates the processing of the D-loop."

We thank the Reviewer for this suggestion. We have tested the effect of the TRR complex on the D-loop processing activity of BLM. Indeed, the interaction partners markedly altered the behavior of the helicase. We have inserted these new data into the current version and completely rewrote the Discussion.

"7. The manuscript is very well written, there is one tiny spelling mistake: Page 16 of the manuscript, last paragraph, line 2, "DNA strand" is misspelt as "DNA stand".

We have corrected this typo.

Reviewer #2 (Remarks to the Author):

"BLM helicase and its homologs in model systems such as Sgs1 in the budding yeast play multiple roles during homologous recombination. Both are involved in the initial resection of the double-strand break in conjunction with the DNA2 nuclease activity. BLM has also been shown to destabilize RAD51-ssDNA filaments under certain biochemical conditions, and the physiological relevance of this activity remains unproven. Moreover, BLM and Sgs1 have been shown to dissociate protein-free D-loop, the product of homology search and DNA strand invasion. Furthermore, from genetic observation during meiotic recombination in budding yeast, it was inferred that Sgs1 stabilizes the crossover dedicated Single-End Invasion intermediate. Finally, BLM and its complex partners TOPOIIIalpha-RMI1-RMI2, as the human yeast homologous complex Sgs1-Top3-Rmi1, can process double Holliday junction to non-crossover products in a reaction that requires both the helicase and topoisomerase (decatenase) activity of the complex.

The present manuscript utilizes purified BLM wild type and mutant constructs on short, oligonucleotide-based substrates to dissect the activities of BLM on D-loops. The authors conclude from the in vitro observations that BLM balance D-loop formation and disruption in vivo.

The fundamental problem is that BLM does not work in isolation. First, it functions in complex with TOPOIIIalpha-RMI1-RMI2."

In the new, significantly extended and developed version of the work we have comprehensively elucidated the effect of TOPOIIIalpha-RMI1-RMI2 (TRR) on BLM's D-loop processing activity, and the BTRR mechanism has become a major focus of the article.

"Second, the enzyme functions in the context of the HR machinery RAD51-RAD54 and other proteins which decorate the DNA intermediates during HR. In fact, for the yeast Sgs1 this has been elaborated, and it is disappointing that this relevant work was not considered. While Sgs1 dissociates protein-free D-loops like BLM, it cannot dissociate D-loops in reconstituted reactions with Rad51, Rad54, RPA in isolation. Rather under reconstituted conditions, the topoisomerase activity of Top3-Rmi1 or the STR complex actively dissociates D-loops (Fasching et al. 2015 Mol. Cell). These in vitro observations are congruent with in vivo data showing that D-loop intermediates accumulate in Sgs1-deficient cells, but this accumulation is independent of its ATPase (i.e. helicase activity) and only dependent on the Top3 decatenase activity (Piazza et al. 2017 Cell 2017, Piazza et al. 2019 Mol. Cell). While the authors selectively discuss work with yeast Sgs1 to support their conclusions, they do not discuss the relevant work done in vivo and in vitro with the yeast enzymes cited above."

We do agree that the mentioned seminal works on the yeast STR complex (and its constituent proteins) constitute an important context for the current findings, and we have included their discussion in the text (Discussion, p. 8., refs. 15, 16 and 20). Nonetheless, our results also reveal intriguing mechanistic differences between the yeast and human systems. We find that human TRR has only a limited effect on the stability of the assessed D-loop constructs (Fig. S10).

According to the new results, we have significantly rewritten both the Introduction and Discussion sections, accounting for the context of the previous findings on Sgs1 and other BLM homologs.

“The present conclusions severely muddle the field and the paper should not be published. The biochemical characterization of BLM on protein-free substrates adds biochemical information, but the results will likely bear little biological significance. Hence submission to a more specialized journal deems more appropriate, which would still require significant rewriting of conclusions and interpretations as well as discussions of the prior work in yeast.”

The deciphered D-loop disruption mechanisms for BLM and BTRR, under the applied conditions, bear far-reaching implications for various biological contexts, as exemplified below.

(i) While BLM indeed appears unable to disrupt active RAD51 bound strand invasions *in vitro*, its D-loop-regulating activity in cells is evidenced by the increase in the frequency of aberrant joint DNA molecules in the absence of BLM in humans or Sgs1 in yeast (refs. 22,23; Discussion, p. 8). BTRR's D-loop disruption activity may aid elimination of a subset of nascent strand invasions on which recombinases are already inactivated, thus contributing to the suppression of non-allelic recombination events, as shown recently for *C. elegans* Him-6 helicase, a BLM ortholog (ref. 17; Discussion, p. 8).

(ii) Besides D-loop disruption, our results suggest that stabilization/extension of D-loops by helicases, a rarely investigated activity, should also be considered in cellular models of HR. For instance, during meiosis, BLM and its orthologs have been shown necessary to ensure HR progression toward crossover formation at designated sites, while inhibiting formation of crossovers at non-designated loci (refs. 4,23; Discussion, p. 8). Thus, BLM alone (or in complex with other factors) may stabilize and extend D-loops at crossover-designated sites and this activity can lead to the subsequent formation of a HJ structure (Fig. 9). Similar scenarios may occur in somatic cells, too, with different further steps channeling the reaction toward non-crossover outcomes (Discussion, p. 9).

(iii) Our study also represents a comprehensive mechanistic elucidation of the phenomenon whereby HR factors significantly influence the outcome of DNA processing reactions by helicases. It is highly likely that, in the near future, further interaction partners of BLM and other helicases will be shown to influence and regulate DNA processing reactions in a manner that is critical for biological outcomes.

REVIEWERS' COMMENTS

Reviewer #1 (Remarks to the Author):

The authors have done an excellent job in their revised ms.
I have no further comments.

Reviewer #2 (Remarks to the Author):

Harami et al. revised

Intrinsic balance between D-loop disruption and stabilization by human Bloom's syndrome helicase"

This revised version does little to address my concern about the original manuscript that the results with short protein-free oligonucleotide-based substrates bear little relevance for the *in vivo* role of the BTRR complex. While the authors now discuss the relevant literature of the yeast Sgs1-Top3-Rmi1 complex, they do not address this concern experimentally. In their discussion of the physiological relevance the authors continue to cherry pick and misrepresent the published record for their purposes. Reference 23 is a New and Views of three articles on yeast STR, where the *in vivo* published work (see<https://pubmed.ncbi.nlm.nih.gov/25699707/>) shows that Top3 is the relevant catalytic activity not Sgs1. Sgs1 serves a more structural role in that context. The discussion on D-loop extension (and these parts of the model figures) has no connection to the present manuscript, as there are no experiments involving D-loop extension.

Reviewer #3 (Remarks to the Author):

BLM is a 3'–5' RecQ DNA helicase involved in the repair of DNA double-strand breaks by the homologous recombination (HR) pathway. In addition to promoting dsDNA end-resection at the site of damage, BLM has been shown to possess also anti-recombinogenic activities. One of such activities is the disruption of displacement loops (D-loops) formed upon invasion of the resected ssDNA into the homologous DNA molecule. The BLM-interacting protein complex Topo III α /RMI1/RMI2 (TRR) plays a role in end resection and likely also in D-loop displacement, but the molecular mechanism by which BLM and TRR cooperate to process D-loops is not completely understood.

In previous work, Harami et. al. developed a kinetic assay to study D-loops processing by the RecQ helicase, based on separately monitoring the processing of 3 structures (four-stranded and three-stranded D-loop-loke structures and a dsDNA structure) using fluorescently labeled oligonucleotides and gel electrophoresis, followed by global fitting to a kinetic model. In this work, the authors take advantage of this same approach to characterize the activity of BLM and TRR. Using the full-length helicase, as well as truncation constructs, their results show that the accessory domains of BLM are important to maintaining the balance between D-loop disruption and extension. Moreover, they show that the TRR proteins shift the equilibrium between these activities towards efficient D-loop disruption.

Overall, this is an interesting and timely work. The data and the analysis are of high quality and the paper is clearly written. The important comments made by the reviewers in the previous submission were fully addressed, and further improved the quality of this manuscript.

This reviewer does not believe that additional revisions will have a significant impact on the paper, and I recommend accepting the paper as is.

Response to reviewer comments

Reviewer #1 (Remarks to the Author):

“The authors have done an excellent job in their revised ms. I have no further comments.”

We thank the reviewer for the positive evaluation.

Reviewer #2 (Remarks to the Author):

*“Harami et al. revised
Intrinsic balance between D-loop disruption and stabilization by human Bloom’s syndrome helicase”*

This revised version does little to address my concern about the original manuscript that the results with short protein-free oligonucleotide-based substrates bear little relevance for the in vivo role of the BTRR complex. While the authors now discuss the relevant literature of the yeast Sgs1-Top3-Rmi1 complex, they do not address this concern experimentally. In their discussion of the physiological relevance the authors continue to cherry pick and misrepresent the published record for their purposes. Reference 23 is a New and Views of three articles on yeast STR, where the in vivo published work (see <https://pubmed.ncbi.nlm.nih.gov/25699707/>) shows that Top3 is the relevant catalytic activity not Sgs1. Sgs1 serves a more structural role in that context. The discussion on D-loop extension (and these parts of the model figures) has no connection to the present manuscript, as there are no experiments involving D-loop extension.”

We are grateful for the critical comments as the points raised by the reviewer helped to improve the manuscript and to shape our interpretation. Below we provide a detailed response to the mentioned points. The relevant key findings and implications are now contained in the significantly rewritten Discussion.

Oligonucleotide-based D-loop structures are indeed different from strand invasion that form in vivo in that in vivo strand invasions are longer, are likely coated with various proteins in different states, and are likely under topological constraints. However, short model D-loops retain important structural features, i.e., the presence of a strand invasion and junction accompanied by a displaced ssDNA strand. Thus, such model systems convey utility in determining the modes of interaction between the D-loop and processing enzymes that are dictated by these features and in testing how protein structural elements and/or binding partners influence these interaction patterns. Indeed, while our system has the above-mentioned limitations, it is especially useful for quantifying all conceivable D-loop binding orientations and their associated processing pathways.

Importantly, our experiments revealed that BLM alone can unwind the double-stranded DNA next to the junction of the strand invasion, analogous to a “recipient” DNA molecule. In line with this, Bachrati et al. (ref. 13 in the manuscript) observed similar behavior using multi-round enzyme kinetic assays together with a similar short model D-loop structure and concluded that BLM may unwind the “recipient” dsDNA parts in their experiment. However, the precise characterization of the process was not performed; the authors instead focused on disruption of plasmid-based mobile D-loops. It must be noted that the only detectable product of such plasmid-based D-loop reactions is the labeled invading strand; thus, possible binding orientations remain unresolved. We found that the activity of BLM in our experiments is brought about by the interplay of various protein structural elements and unwinding of recipient dsDNA is apparently inhibited by the TRR complex. In contrast, the BTRR complex is an efficient

disruptor of the model D-loop structure via elimination of the strand invasion and the BTRR complex interacts in such a mode with the D-loop that is in line with previous assumptions.

Monitoring these features on more complex D-loop structures having longer invasions and a complementary displaced strand, containing topological strain and/or bound by a multitude of proteins would be technically highly challenging. The limited in vitro processivity of single helicase runs, complications arising from dynamic fluctuations brought about by topological strain and bound proteins, combined with the detection limits of fluorescence methods, would make detection of reaction intermediates and products uncertain or even impossible under single-round kinetic conditions, which are required for dissection of pathways. Also, the effect of TopoIII α was stated to be non-measurable (Bachrati et al., ref. 13), as TopoIII α alone induced the release of the invading strand from the plasmid-based D-loop. The authors stated that this activity was observed due to TopoIII α cutting of the plasmid backbone based on the observation that a simple restriction digest caused the same effect (Bachrati et al., ref. 13). However, Fasching et al. (ref. 20) showed no relaxation activity for human TRR on intact plasmids and relaxation activity was only observed on D-loop containing plasmids and inferred that TRR activity is specialized for the D-loop.

It must also be noted that most of the mechanistic conclusions for the activity profiles of RecQ helicases have been derived using reductionist in vitro model systems. The molecular activities of RecQ helicases unwinding protein-free D-loops (van Brabant et al., ref. 14) and R-loops (Chang et al. 2017 J Cell Biol), dissolving Holliday junctions (Wu and Hickson, ref. 25), unwinding G-quadruplex structures (Sun et al. 1998 J Biol Chem) or acting on branched DNA substrates (Mohaghegh et al. 2001 Nucleic Acids Res) were all established using short model DNA structures.

The precise role of BLM homologs in somatic and meiotic HR apparently varies between different model organisms. However, BLM and its homologs are generally identified as essential regulators of HR outcome, together with TopoIII and RMI proteins. As pointed out by the reviewer, extensive studies have been published using the *S. cerevisiae* model system. The paper mentioned by the reviewer was (and is) referred to in the manuscript (Kaur et al., ref. 55 in the revised version), and we did emphasize both in the Introduction (page 2) and Discussion (page 8) that in yeast the catalytic activity of the Top3-Rmi1 complex is required for D-loop disruption whereas the helicase function of Sgs1 seems to be largely dispensable based on the results of Piazza et al. (ref. 15) and Fasching et al. (ref. 20). However, it is noteworthy that the increased crossover frequency in Δ Sgs1 background is not complemented by helicase-dead Sgs1 (Ira et al. 2003 Cell, ref 54 in the revised version). The helicase function was found important only for some STR activities by Weinstein & Rothstein (2008 DNA repair); and Lo et al. (2006 Mol Cell Biol, ref 60 in the revised version) showed helicase function-independent regulation of crossovers. While Jain et al. (2009 Genes & Dev) monitoring BIR repair found that Sgs1 can complement the Δ Sgs1 phenotype, a helicase-dead mutant impeded repair, instead of complementing the phenotype. Tang et al. (ref. 56 in the revised version) also reported a helicase function-dependent role for STR in HR, albeit both Tang et al. and Kaur et al. (ref. 55) also reported a late Sgs1-independent function for TR in the resolution of entangled chromosomes. In addition, helicase-dead BLM mutants are unable to complement the increased SCE phenotype of Bloom's syndrome cells (Neff et al. 1999 Mol Cell Biol). Importantly, the contributions of various presumed functions mediated by the mentioned enzymes to resection and DHJ dissolution were not separated in the aforementioned in vivo studies, except for Piazza et al. (ref. 15) where D-loop dynamics were specifically monitored. Based on these observations, yeast Sgs1 emerges as a structural protein enhancing the activity of the STR complex towards D-loop disruption without a necessity for processive Sgs1-driven DNA unwinding, whereas DHJ dissolution appears to require robust helicase activity. In addition, helicase-dead Sgs1 may well interact with D-loops (as paralleled by strong binding of BLM to D-loops in the absence of ATP observed in our work) and passively melt a region in the structure, which in turn can serve as a substrate for TR activity. Passive DNA melting based on structural considerations has been proposed for BLM by Kitano (2014 Front Genetics). Importantly, the possibility of different D-loop binding orientations was not considered in previous studies.

Our proposed activity of BLM in HR, which includes both TRR-independent and TRR-dependent functions, is in line with *in vivo* findings in higher eukaryotes. While yeast genetic studies suggest that most functions of Sgs1 and the TR complex are strictly linked (Tang et al., ref. 56), the *C. elegans* homolog of RMI1 strictly colocalizes with Him-6 (BLM homolog) only during late pachytene but not in mid-pachytene (Jagut et al. 2016 Plos Biology). Similarly, BLM and TopoIII α colocalize only partially on meiotic human chromosomes (Johnson et al., ref. 58). While the presence of active recombinase at the strand invasion apparently inhibits D-loop disruption by BLM (Bugreev et al., ref. 11), our results imply that the unwinding of the recipient DNA by BLM could stabilize the D-loop and we propose that this activity may migrate the branch point away from the initial strand invasion. In an *in vivo* context, the recipient DNA close to the formed strand invasion is not recombinase-bound and, thus, the unwinding of this DNA segment by BLM could facilitate the escape of a subset of nascent D-loops from disruption. Hence, BLM's activities described in our current work could be relevant for HR regulation. In the new version we uniformly denote the unwinding of the "recipient" dsDNA segment as "stabilization" (as opposed to "extension" in the previous version) to unambiguously delineate this activity from polymerase-mediated D-loop extension.

Reviewer #3 (Remarks to the Author):

"BLM is a 3'–5' RecQ DNA helicase involved in the repair of DNA double-strand breaks by the homologous recombination (HR) pathway. In addition to promoting dsDNA end-resection at the site of damage, BLM has been shown to possess also anti-recombinogenic activities. One of such activities is the disruption of displacement loops (D-loops) formed upon invasion of the resected ssDNA into the homologous DNA molecule. The BLM-interacting protein complex Topo III α /RMI1/RMI2 (TRR) plays a role in end resection and likely also in D-loop displacement, but the molecular mechanism by which BLM and TRR cooperate to process D-loops is not completely understood.

In previous work, Harami et. al. developed a kinetic assay to study D-loops processing by the RecQ helicase, based on separately monitoring the processing of 3 structures (four-stranded and three-stranded D-loop-loke structures and a dsDNA structure) using fluorescently labeled oligonucleotides and gel electrophoresis, followed by global fitting to a kinetic model. In this work, the authors take advantage of this same approach to characterize the activity of BLM and TRR. Using the full-length helicase, as well as truncation constructs, their results show that the accessory domains of BLM are important to maintaining the balance between D-loop disruption and extension. Moreover, they show that the TRR proteins shift the equilibrium between these activities towards efficient D-loop disruption.

Overall, this is an interesting and timely work. The data and the analysis are of high quality and the paper is clearly written. The important comments made by the reviewers in the previous submission were fully addressed, and further improved the quality of this manuscript.

This reviewer does not believe that additional revisions will have a significant impact on the paper, and I recommend accepting the paper as is."

We thank the reviewer for the positive evaluation of our work.